# Postnatal oogenesis leads to an exceptionally large ovarian reserve in naked mole-rats

Miguel Angel Brieño-Enríquez [1,2] ✉, Mariela Faykoo-Martinez[3,4,5], Meagan Goben[1], Jennifer K. Grenier[6], Ashley McGrath[7], Alexandra M. Prado[7], Jacob Sinopoli[7], Kate Wagner[7], Patrick T. Walsh[1], Samia H. Lopa[1], Diana J. Laird [8], Paula E. Cohen[9], Michael D. Wilson [5,10], Melissa M. Holmes[3,4] & Ned J. Place [7] ✉

In the long-lived naked mole-rat (NMR), the entire process of oogenesis occurs postnatally. Germ cell numbers increase significantly in NMRs between postnatal days 5 (P5) and P8, and germs cells positive for proliferation markers (Ki-67, pHH3) are present at least until P90. Using pluripotency markers (SOX2 and OCT4) and the primordial germ cell (PGC) marker BLIMP1, we show that PGCs persist up to P90 alongside germ cells in all stages of female differentiation and undergo mitosis both in vivo and in vitro. We identified VASA+ SOX2+ cells at 6 months and at 3-years in subordinate and reproductively activated females. Reproductive activation was associated with proliferation of VASA+ SOX2+ cells. Collectively, our results suggest that highly desynchronized germ cell development and the maintenance of a small population of PGCs that can expand upon reproductive activation are unique strategies that could help to maintain the NMR's ovarian reserve for its 30-year reproductive lifespan.

Female reproductive senescence in mammals largely results from the depletion of a finite number of ovarian follicles that were produced during embryonic development or early postnatal life[1]. To date, the vast majority of research on factors that influence the establishment and maintenance of the ovarian reserve has been performed in mice, which have relatively short reproductive lifespans, measured in terms of months. In mouse and human ovaries, the primordial follicles, each of which contains an oocyte surrounded by a single layer of somatic pre-granulosa cells, represent the entire ovarian reserve, which will supply oocytes throughout the reproductive lifespan[2]. In mice and humans, these follicles are produced from a pool of primordial germ cells (PGCs), which localize to the developing gonads during gestation[1,3–5]. Mitotic division of PGCs is characterized by incomplete cytokinesis, producing clusters of interconnected oogonia, and forming germ cell nests. Thereafter, the oogonia enter into meiotic prophase I, at which time they are referred to as oocytes and remain arrested in this stage of meiosis. Germ cell nests then begin to breakdown, during which the majority of oocytes are lost through apoptotic cell death, whereas the remaining oocytes become surrounded by a layer of somatic pre-granulosa cells, forming primordial follicles[6–8]. This generally occurs during mid-to-late gestation in the mammalian ovary and establishes the ovarian reserve at birth (human) or shortly thereafter (mice)[9,10]. Thus, a key determinant of the ovarian reserve size is the total number and quality of the germ cells in the ovary[11].

[1]Magee-Womens Research Institute, Department of Obstetrics, Gynecology & Reproductive Sciences, University of Pittsburgh, Pittsburgh, PA, USA. [2]Aging Institute, University of Pittsburgh School of Medicine, Pittsburgh, PA, USA. [3]Department of Cell and Systems Biology, University of Toronto, Toronto, ON, Canada. [4]Department of Psychology, University of Toronto, Mississauga, Mississauga, ON, Canada. [5]Program in Genetics and Genome Biology, The Hospital for Sick Children, Toronto, ON, Canada. [6]RNA sequencing core and Center for Reproductive Genomics, College of Veterinary, Cornell University, Ithaca, NY, USA. [7]Department of Population Medicine & Diagnostic Sciences, Cornell University, Ithaca, NY, USA. [8]Department of Obstetrics, Gynecology & Reproductive Sciences, University of California San Francisco, San Francisco, CA, USA. [9]Center for Reproductive Genomics, Department of Biomedical Sciences, Cornell University, Ithaca, NY, USA. [10]Department of Molecular Genetics, University of Toronto, Toronto, ON, Canada. ✉e-mail: brienoenriquezma@mwri.magee.edu ; njp27@cornell.edu

The naked mole-rat (NMR, *Heterocephalus glaber*) is well known for being the longest-lived rodent, with a maximum lifespan of >37 years[12–14], which is 368% longer than predicted based on body mass[15]. They are also highly resistant to cancer and prolonged anoxia and are one of only two eusocial species of mammal[16–20]. Female NMRs are also exceptional because they demonstrate no decline in fertility or fecundity for the vast majority, if not the entirety of their decades-long lifespan[13,21]. Conversely, comparably sized mice have reduced fertility by 9 months of age[11]. Almost all NMR colonies have a single dominant, breeding female (queen), and she will continue to breed until she dies[22]. All other females are subordinate to the queen and most likely will never become reproductively mature[16]. The queen's protracted fertility is very relevant since the functional lifespan of the ovary hinges upon the finite number of oocytes that progressively decreases with age. While the histological images of NMR ovaries reported by Kayanja and Jarvis suggested an unusually large number of primordial follicles[23], at 6 months of age the NMR ovarian reserve is more than 10-fold larger than in mammals of a comparable size[24]. However, the mechanism(s) by which the NMR establishes this large ovarian reserve during fetal and perinatal life and the mechanism(s) allowing them to avoid reproductive senescence remain unknown.

In this study, we show that the NMR ovary retains a population of germ cells far into postnatal life that bear markers of pluripotency as well as PGC identity and are capable of dividing both in vivo and in vitro. Cells with the hallmarks of post-migratory PGCs (VASA and SOX2) were abundant in the ovaries of subordinate females at 6 months and 3 years of age and were proliferative in the ovaries of 3-yr-old reproductively activated adults. We also demonstrate the co-existence of germ cells at every major stage of development in the postnatal NMR ovary (from early postnatal life up to 3 years of age), including pre-meiotic cells, cells at meiotic prophase I and follicles, which is similar to what is described in human fetal ovaries[25,26]. Collectively, we establish that the entire process of oogenesis occurs postnatally in NMR ovaries, which provides strong support for the hypothesis that pre-meiotic germ cell proliferation throughout the postnatal period generates an exceptionally large ovarian reserve relative to the NMR's body size.

## Results

### Establishment of NMR ovarian reserve

To understand whether NMRs establish an ovarian reserve sufficient for escaping reproductive senescence, we performed histological analyses on NMR ovaries at postnatal (P) days 1, 5, 8, 15, 28, and 90 (Fig. 1a, b). Hematoxylin and eosin (H&E) staining revealed a large number of germ cells at all ages analyzed. In mice, the entire population of PGCs engages in the oogenesis program at embryonic (E) day 13.5, and meiotic prophase I initiates thereafter[27]. Following meiotic initiation, oocytes form nests that later breakdown during primordial follicle formation, a process completed in mice by P5[6,7,28]. In NMR ovaries, we did not observe any primordial follicles at P1 and P5, and the first primordial follicles were observed within the ovarian medulla at P8. Primary follicles (early growing follicles with a single layer of cuboidal granulosa cells) were also first identified at P8. As late as P90, mixed populations of germ cells including germ cell nests, primordial, primary, and secondary follicles were observed (Fig. 1a). Histological analyses showed that in the NMR the number of germ cells present in the ovary is exceptionally large relative to other comparably sized rodents. To verify their germ cell lineage, we performed immunofluorescence (IF) on paraffin-embedded sections at different ages using antibodies against VASA (DDX4). In mouse, *Vasa* expression is first observed in late migratory PGCs and continues into meiotic prophase I and folliculogenesis[27,29]. Immunofluorescence for VASA revealed a remarkable number of germ cells in NMR ovaries from P1 to P90 (Fig. 1b). We quantified the number of germ cells in histological sections from six different animals at six different ages following a

previously published algorithm[24,30]. Our results show that germ cell numbers in NMRs significantly increase between P5 and P8, with 1.5 million germ cells per ovary pair at P8 (Fig. 1c). The number of germ cells at P8 was statistically greater than all other ages examined, raising the possibility that germ cells are actively dividing in the postnatal ovary of the NMR. To test this hypothesis, we performed IF with antibodies against VASA and two different proliferation markers: proliferation marker protein Ki67 (Ki67)[31] and the phosphorylated histone H3 (pHH3; phosphorylation on serine-10)[32,33]. We observed VASA+ Ki67+ cells in ovaries from P1 to P90 (Fig. 1d). At P1, P5 and P8, these cells formed clusters, in contrast to dyads at P15 and P28, and single, isolated VASA+ Ki67+ cells at P90 (Fig. 1d). A similar distribution of cells along the different ages was observed for VASA+ pHH3+ (Supplementary Fig. 1a). We quantified the numbers of VASA+ Ki67+ and VASA+ pHH3+ cells to understand the effects of age (Fig. 1e and Supplementary Fig. 1b). We observed the maximum number of VASA+ Ki67+ and VASA+ pHH3+ cells at P1 followed by a reduction at P5 and a new peak at P8, which coincides with the highest number of VASA+ cells (Fig. 1c). Although the numbers of VASA+ Ki67+ and VASA+ pHH3+ cells declined after the P8 peak, they remained present at 3 months of age (P90).

### Presence of PGCs in postnatal NMR ovaries

Germ cells express *Vasa* at different stages, including PGCs, gonocytes, and oocytes in meiotic prophase I[27,34]. As the embryonic precursors of the germ cell lineage, PGC precursors in mice are specified in the epiblast around 6.25 days postcoitum (dpc)[27,35,36]. PGC specification depends on the key factors *Blimp1* (*Prdm1*)[37], *Tfap2c*[38], and *Prdm14*[39], which repress the somatic program and promote epigenetic reprogramming and re-expression of pluripotency transcription factors (TFs), such as *Sox2* and *Oct4* (*Pou5f1*)[27,40]. SOX2 and OCT4 are key players in a self-reinforcing network of TFs that maintain self-renewal and pluripotent identities by regulating the expression of pluripotency genes and suppressing genes that stimulate differentiation[40]. To characterize the specific germ cell populations in the NMR ovary, we tested antibodies against SOX2, OCT4, and BLIMP1 on fetal ovary sections at E56 (Supplementary Fig. 2), in addition to postnatal NMR ovaries. We identified germ cells positive for VASA and SOX2 at every age (Fig. 2a). To quantify the total number of VASA+ SOX2+ cells at each postnatal age, we followed the same strategy used to enumerate the total number of germ cells. Quantification across ages revealed that the greatest number of VASA+ SOX2+ cells was at P5 (Fig. 2b). The number of VASA+ SOX2+ cells declined significantly from P8 to P15, and by P90 comprised just 0.03% of the total number of germ cells. Because the expression of OCT4 is also crucial for PGC programming, we analyzed the number of VASA+ OCT4+ cells at different ages (Fig. 2c). The largest numbers of VASA+ OCT4+ cells were observed from P1 to P8 (Fig. 2d). A decline in the total number of cells was observed from P15 to P90, but VASA+ OCT4+ cells were still present at 3 months of age. Murine PGCs and pluripotent stem cells are positive for both SOX2 and OCT4, however only PGCs express BLIMP1. To differentiate PGCs from pluripotent stem cells within the NMR ovary, we stained with BLIMP1. Immunofluorescence analysis of NMR ovaries showed the presence of VASA+ BLIMP1+ cells at all ages between P1 and P90 (Fig. 2e). Similar to the pattern for VASA+ OCT4+, VASA+ BLIMP1+ cells were enriched at early postnatal ages (P1-P8), followed by reductions that led to the lowest level at P90 (Fig. 2f). We observed that the number of VASA+ SOX2+ cells was greater than the number of VASA+ OCT4+ cells, and therefore, we co-stained ovaries for SOX2 with OCT4 (Fig. 2g). We observed that the number of cells positive for both proteins was very similar to the total number of VASA+ OCT4+ cells (Fig. 2h), indicating that almost all VASA+ OCT4+ cells are positive for SOX2+. Conversely, not all SOX2+ cells are positive for OCT4. These results suggest that in the NMR the SOX2 pluripotency factor might play an essential role in the postnatal maintenance of PGCs.

## In vivo and in vitro mitotic expansion of NMR PGCs

A key characteristic of PGCs is the capacity to mitotically expand and populate the developing gonads[1,41,42]. To determine whether VASA+ SOX2+ cells in the postnatal NMR ovary are mitotically active, we immunolabeled for SOX2 and Ki67. We detected dual-labeled cells for both proteins in ovaries from all analyzed ages (P1 to P90) (Fig. 3a). SOX2+ Ki67+ cells were observed forming clusters in ovaries from P1 to

P8. The numbers of SOX2+ Ki67+ cells were reduced at P15 and P28, where they existed in pairs and as single cells at P90 (Fig. 3a). We quantified the total numbers of SOX2+ Ki67+ cells per animal at all ages (Fig. 3b), and ovaries at P5 had the greatest number of cells positive for both markers, which was also the case for VASA+ SOX2+ (Fig. 2b). The number of SOX2+ Ki67+ cells declined from P8 to P90, but they were still present at 3 months of age. Following this indication that VASA+

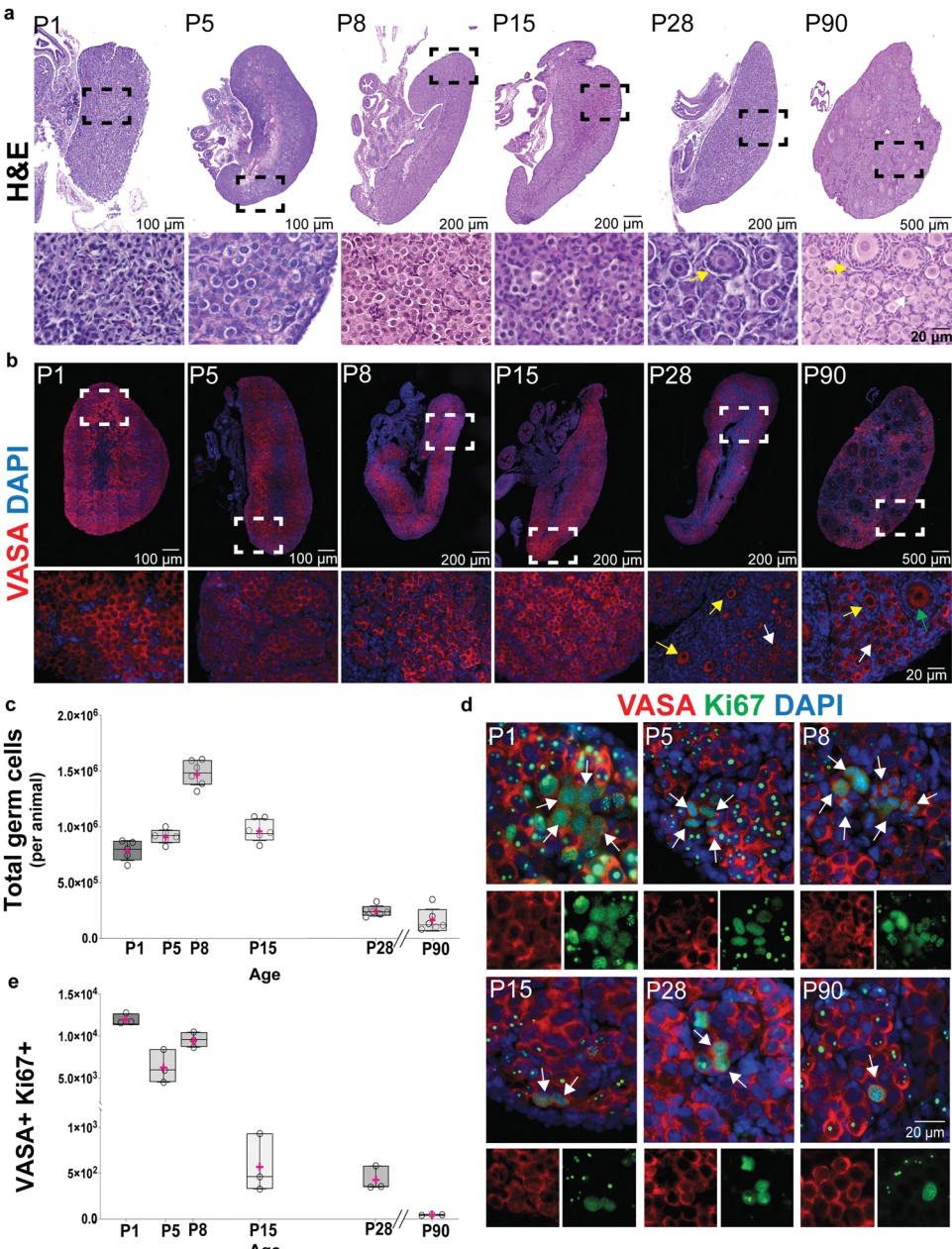

**Fig. 1 | Postnatal ovaries from naked mole-rats (NMR) have a large ovarian reserve with actively dividing germ cells. a** Histological sections of NMR ovaries at different stages of development (postnatal day P1, P5, P8, P15, P28, and P90) stained with hematoxylin and eosin. **b** Histological sections stained with antibody against VASA (DDX4; red) and counterstained with DAPI (blue) to reveal DNA. White arrows indicate an example of primordial follicles, yellow arrows indicate an example of primary follicles, and green arrow indicates a secondary follicle. **c** Quantification of the total germ cell numbers per NMR detected with VASA at P1, P5, P8, P15, P28, and P90 (n = 6 per group). In all the graphs the box edges represent 25th and 75th percentiles, the horizontal line inside the box represents the median, and the pink '+' represents the mean. In one-way-ANOVA, adjusting for multiple

comparisons using the Bonferroni method, all pairwise comparisons were statistically significant at p < 0.0001, except for the comparisons P1 vs P5 (p = 0.35), P5 vs P15 (p > 0.99) and P28 vs P90 (p > 0.99). **d** In vivo postnatal germ cell expansion analyzed using immunofluorescence detection of the marker of mitotic cell division Ki67 (green) and the germ cell marker VASA (red). White arrows indicate cells that are positive for both markers. **e** Quantification of the mitotically active germ cells across the different ages using Ki67 and VASA as markers at P1, P5, P8, P15, P28, and P90 (n = 3 per group). For box plots of VASA+ Ki67+, all pairwise comparisons were statistically significant at p < 0.0001, except for age P1 vs P8 (p = 0.11), P15 vs P28, P15 vs P90, and P28 vs P90 (all p > 0.99). Source data are provided as a Source Data File.

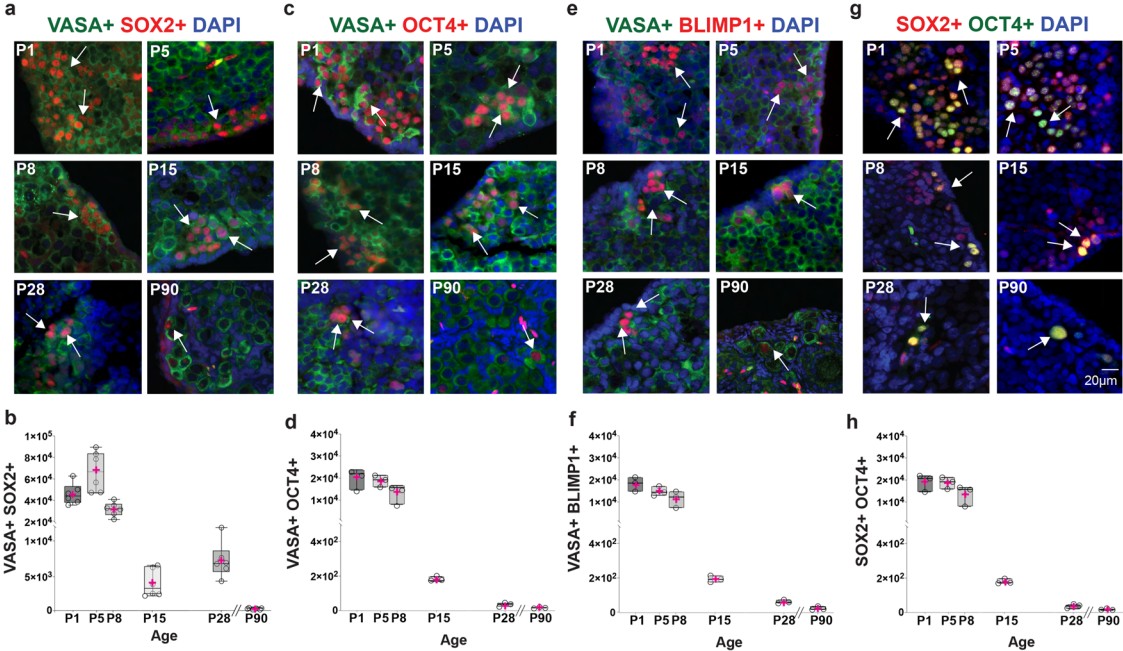

**Fig. 2 | Presence of pluripotency and primordial germ cell markers in postnatal naked mole-rat (NMR) ovaries. a** NMR ovary sections stained with the pluripotency marker SOX2 (red), VASA (green), and DAPI (blue). White arrows indicate representative SOX2 positive cells. **b** Quantification of SOX2 positive cells at P1, P5, P8, P15, P28, and P90 (*n* = 6 per group). In all the graphs, the box edges represent 25th and 75th percentiles, the horizontal line inside the box represents the median, and the pink '+' represents the mean. In one-way-ANOVA, adjusting for multiple comparisons using the Bonferroni method, all pairwise comparisons were statistically significant at *p* < 0.0001, except for P1 vs P8 (*p* = 0.18), P15 vs P28, P28 vs P90, and P15 vs P90 (all *p* > 0.99). **c** NMR ovary sections stained with the pluripotency marker OCT4 (red), VASA (green), and DAPI (blue). White arrows indicate representative SOX2-positive cells. **d** Quantification of VASA+ OCT4+ positive cells at P1,

P5, P8, P15, P28, and P90 (*n* = 3 per group), and all pairwise comparisons were statistically significant at *p* < 0.0001, except for P5 vs P8 (*p* = 0.10), P1 vs. P8, P15 vs. P28, P15 vs. P90, and P28 vs P90 (all *p* > 0.99). **e** Detection of the PGC marker BLIMP1+ (red) and VASA (green) on the NMR ovary. **f** Quantification of VASA+ BLIMP1+ cells at P1, P5, P8, P15, P28, and P90 (*n* = 3 per group), and all pairwise comparisons were statistically significant at *p* < 0.0001, except for P1 vs. P5, P5 vs. P8, P15 vs. P28, P15 vs. P90 and P28 vs. P90 (all *p* > 0.99). **g** Co-staining of the pluripotency markers SOX2+ (red) and OCT4+ (green) on the NMR ovary. **h** Quantification of SOX2+ OCT4+ cells at P1, P5, P8, P15, P28, and P90 (*n* = 3 per group), and all pairwise comparisons were statistically significant at *p* < 0.0001, except for P1 vs. P5, P1 vs. P8, P5 vs. P8, and for those among ages P15, P28, and P90 (all *p* > 0.99).

SOX2+ cells divide in vivo, we tested their capacity to divide in vitro. For this purpose, we cultured small fragments of ovaries (1 × 1 × 1 mm) using a modified version of a technique that has been previously reported for mouse and human ovaries[43,44]. We evaluated ovaries from P1 to P90, but best results were obtained with fragments from P1 and P5 ovaries. Twenty-four hours after seeding P5 ovarian fragments, we observed fibroblasts spreading out from the tissue. Five days later we observed rounded cells with a large nucleolus, and at 10 days of culture these round cells formed clusters that were more evident at days 15 and 20 of culture (Fig. 3c). To probe the lineage of in vitro clusters, after 8 days of culture, we treated P5 ovarian fragments with the thymidine analog 5-ethynyl-2′-deoxyuridine (EdU) for 48 h. The cultures were fixed on day 10 and stained for SOX2, VASA, EdU, and DAPI (Fig. 3d). Our results show all cells in clusters were positive for VASA, of which 48% were VASA+ SOX2+, and 52% were SOX2+ EdU+, indicating that the dividing cells are germ cells. However, not all of them maintain the expression of the pluripotency factor SOX2 (Fig. 3d). Additionally, we extended the culture to 30, 60, 90, and 120 days, during which we observed the progression to the formation of germ cell nests, as well as primordial follicles (Supplementary Fig. 3). Taken together, our results provide evidence that germ cells in postnatal NMR ovaries are capable of dividing both in vivo and in vitro.

### Temporal profile of the NMR transcriptome across ovarian development

To identify molecular pathways involved in NMR ovarian development, we performed RNA sequencing (RNAseq) on ovarian samples collected at E56, P1, P5, P8, P28, and P90. We focused our analysis on

sequential timepoints to identify dynamic shifts in the temporal profile of the transcriptome across ovarian development (Fig. 4, Supplementary Fig. 4, Supplementary Data 1–4). Genes labeled in Fig. 4a plots were selected based on their relevance to oogenesis and reproduction. Gene set enrichment analysis (GSEA) identified dynamic meiotic pathways across timepoints (Fig. 5a, Supplementary Data 5–12). For example, the "meiotic cell cycle" pathway was upregulated between E56 and P1 (NES = 2.18, *p*adj<0.001) and downregulated between P8 and P28 (NES = −2.13, *p*adj<0.001) with no significant differences observed between P1 and P8. These dynamic changes in meiotic gene regulation were also observed using real-time PCR with reverse transcription (RT-qPCR) of meiotic genes *Stra8*, Spo11, *MeioC, Meiob, Ythdc2, Mlh1, Hormad1, Rec8, Dmc1, Sypc1, Sycp3* on independent ovarian samples collected at E56 (*n* = 2), P1 (*n* = 2), P5, P8, P28, and P90 (*n* = 3 per group), We also validated the expression changes of PGC developmental genes, *Sox2, Oct4, Lin28a, Blimp1 (Prdm1), Kit and Tfap2c*. The PGC genes were upregulated from E56 to P8, and then downregulated but still present at P28 and P90 (Fig. 4b; Supplementary Data 1–4). We also completed a time-series differential expression analysis, which identified several genes of interest that were downregulated across age (e.g., *Oct4 (Pou5f1), Top2a*; Supplementary Fig. 5, Supplementary Data 13), although no obvious ovarian development pathways were highlighted by GSEA (Supplementary Data 14, 15).

We next compared meiotic gene regulation during ovarian development between NMRs and mice. To curate a NMR ovarian developmental gene list, we combined the 76 genes identified from meiotic pathways in the GSEA analysis with the 20 genes profiled by RT-qPCR (Supplementary Fig. 6), resulting in 65 genes after

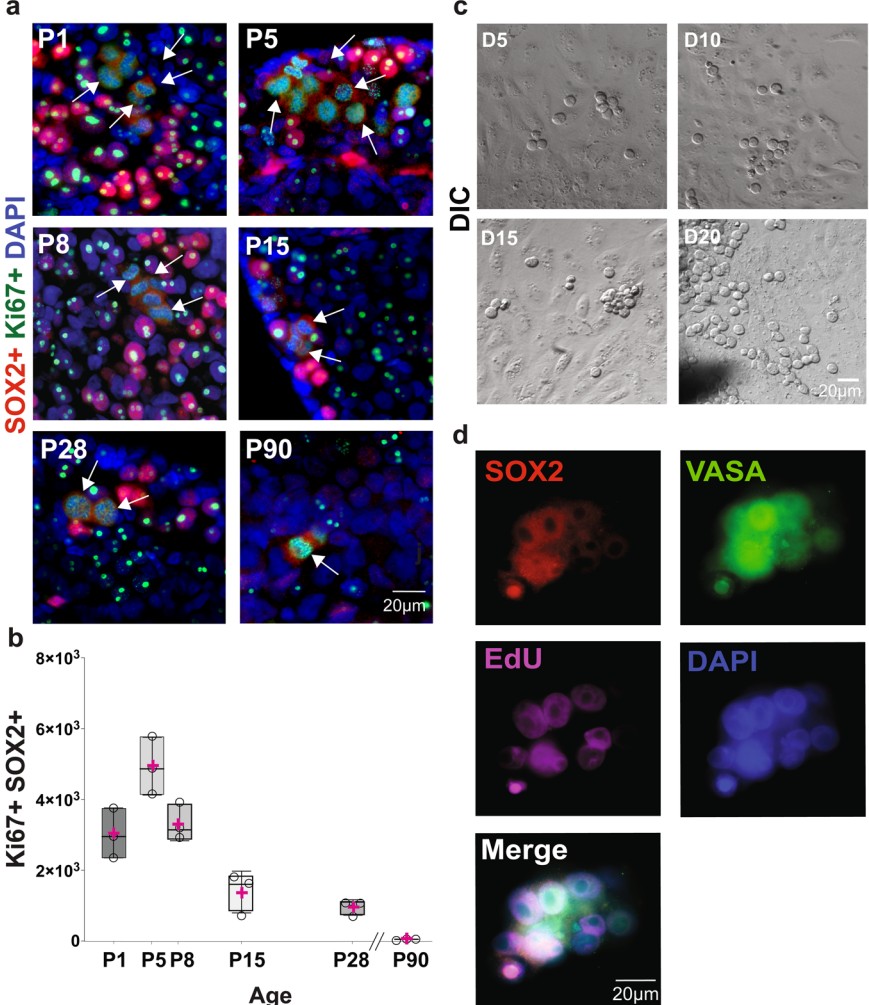

**Fig. 3 | In vivo and in vitro postnatal mitotic germ cell expansion in the NMR ovaries. a** In vivo postnatal SOX2+ cell expansion analyzed using immunofluorescence detection of the marker of mitotic cell division Ki67 (green) and SOX2 (red). White arrows indicate cells that are positive for both markers. **b** Quantification of SOX2+ Ki67+ cells at different ages (P1, P5, P8, P15, P28, and P90; $n = 3$ per group), and all pairwise comparisons were statistically significant at $p < 0.0001$, except for P5 vs P8 ($p = 0.07$), P1 vs P15 ($p = 0.05$), P15 vs P90 ($p = 0.20$), P28 vs P90 ($p = 0.83$), P1 vs P8 and P15 vs P28 (both $p > 0.99$). **c** In vitro expansion of germ cell from the P5 NMR ovary at day D5, D10, D15, and D20 in culture (DIC). **d** Immunodetection of the pluripotency marker SOX2 (red), germ cell marker VASA (green), incorporation of ethynyl-labeled deoxyuridine (EdU) (magenta) and DAPI (blue) in cultured cells from P5 NMR ovary at D10. Source data are provided as a Source Data File.

overlapping genes were removed (Fig. 5b). We compared the expression of NMR meiotic genes to mice at E10.5, E13.5, P3, P14, and P28[45]. Indeed, most meiotic genes exhibited the highest expression at E13.5 in mouse and postnatally in NMR (Fig. 5b, c), highlighting the unusual postnatal upregulation of meiotic genes during postnatal ovarian development in the NMR.

**Initiation of meiotic prophase I in the postnatal NMR ovary**

Murine and human gamete development begins during embryogenesis and is initiated by PGCs[27,46]. To contribute to fertility, PGCs must migrate to the gonad, divide mitotically to populate the gonad, and then proceed through meiotic prophase I, at which time they are referred to as oocytes. Meiotic prophase I is the first and the longest stage of mammalian meiosis, and during this stage homologous chromosomes must complete the processes of pairing-synapsing, and complete meiotic recombination to generate a crossover event. Entry into meiosis is initiated and regulated in part by *Stra8*[47], *MeioC*[48], and *Ythdc2*[49]. To determine the age(s) at which meiosis initiates in NMR ovaries, we performed IF for STRA8 and found few STRA8+ cells at P1 (Supplementary Fig. 7a). We quantified the number of VASA+ STRA8+

cells from P1 to P90. Our results showed the greatest numbers of VASA + STRA8+ cells at P8 to P28, followed by a reduction at P90 (Supplementary Fig. 7b). This result is consistent with RNAseq and RT-qPCR analyses (Fig. 4a, b, and Supplementary Fig. 6). *MeioC* is also involved in meiotic entry and was detected at P1 and continuously up to P90 (Fig. 4a, b, and Supplementary Fig. 6), which resembles the pattern of *Stra8* expression (Fig. 4a, b, and Supplementary Fig. 6).

To demonstrate progression into meiotic prophase I, we stained ovarian sections with the meiosis-specific cohesin REC8 and co-stained with SOX2. Immunofluorescence for REC8 and SOX2 revealed the co-existence of REC8+ and SOX2+ cells in NMR ovaries from P1 to P90 (Fig. 6a). We also evaluated the number of VASA+ REC8+ cells from P1 to P90 (Fig. 6b), and their numbers steadily increased from P1 ($\bar{x} = 130{,}223$ cells per ovary pair) to P8 ($\bar{x} = 1{,}147{,}154$ cells per ovary pair). The total numbers of VASA+ REC8+ cells declined thereafter from P15 ($\bar{x} = 878{,}343$) to P90 ($\bar{x} = 128{,}033$ cells per ovary pair) (Fig. 6b, c). Whereas we observed REC8 signal in follicles as expected, we excluded REC8+ follicles from the counts, because we wanted to quantify only those oocytes during meiotic prophase before the dictyate arrest and follicle

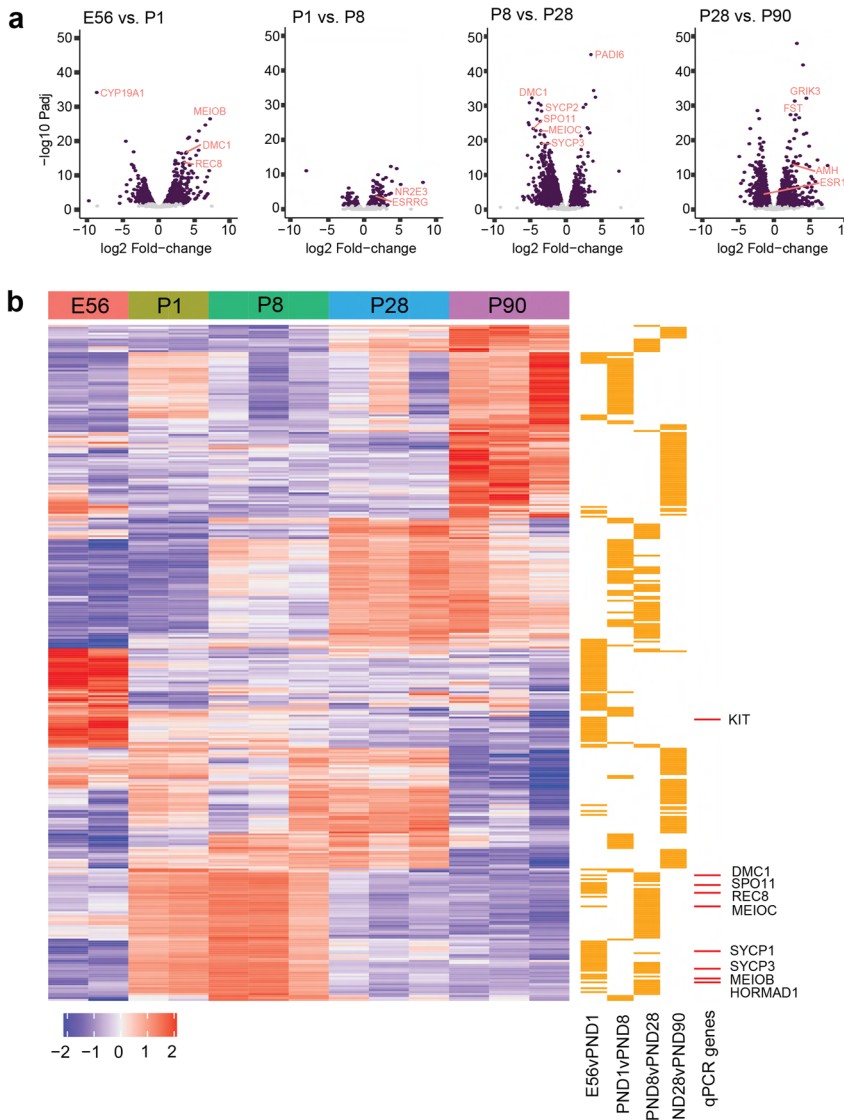

**Fig. 4 | RNAseq analysis of ovaries at E56, P1, P8, P28, and P90. a** Volcano plot with genes that are significantly different coloured in purple (adjusted *p* value <0.05 and log2 fold-change > abs(log2(1.5))). **b** A heatmap of the top 50 upregulated and top 50 downregulated differentially-expressed genes from each sequential pairwise comparison (counts per million). Gene expression is scaled and clustered by unsupervised hierarchical clustering. The orange bars to the right of the heatmap mark from which comparison the gene originates. The red bars to the right of the heatmap in **b** mark which of the DEGs were also profiled in the qPCR experiment. Source data are provided as a Source Data File.

development. This explains, at least in part, the apparent reduction in the numbers of REC8+ germ cells between P15 and P90.

The REC8 signal within histological sections indicates the presence of oocytes at meiotic prophase I, but it does not clearly distinguish between the four different substages (leptonema, zygonema, pachynema, and diplonema) that precede dictyate arrest. Meiotic prophase progression was delineated using prophase I chromosome spreads to detect the two defining features of this stage: synapsis of homologous chromosomes and initiation of meiotic recombination by DNA double-strand break (DSB) formation[50,51]. Meiotic prophase I substages are defined cytologically by the appearance of the synaptonemal complex[50]. Leptonema is the first substage and follows premeiotic DNA replication that results in the sister chromatids being physically linked together via cohesin complexes[52]. At zygonema, the homologous chromosomes pair and begin to synapse as the central element proteins promote this process[53–55]. Once the homologous chromosomes are completely condensed and synapsed, the cell has entered pachynema and proceeds with the last steps of repairing the

programmed DSBs. Immediately after pachynema is completed, proteins from the central element begin to disassemble during diplonema, and homologous chromosomes remain connected at the chiasmata[50,51]. In other mammals, including mouse and human, oocytes remain in dictyate arrest until shortly before ovulation. However, for NMRs, the majority of subordinate females never undergo reproductive activation and maintain their oocytes in dictyate arrest throughout life.

As mentioned above, the different substages of meiotic prophase I are typically evaluated according to the characteristics of the synaptonemal complex. However, as the antibodies for these proteins (SYCP2, SYCP3) used in mice produced inadequate results in NMRs, we used an antibody for the meiotic cohesin REC8. These investigations were performed at P8 and P28 (Fig. 6d–g and Supplementary Fig. 8), and these ages were selected based on the signals observed on tissue sections (Fig. 6b) as well as results from RNAseq and RT-qPCR (Figs. 4, 5 and Supplementary Fig. 6). In P8 and P28 ovaries, we detected all substages of meiotic prophase I (Fig. 6d, e, f). The progression of

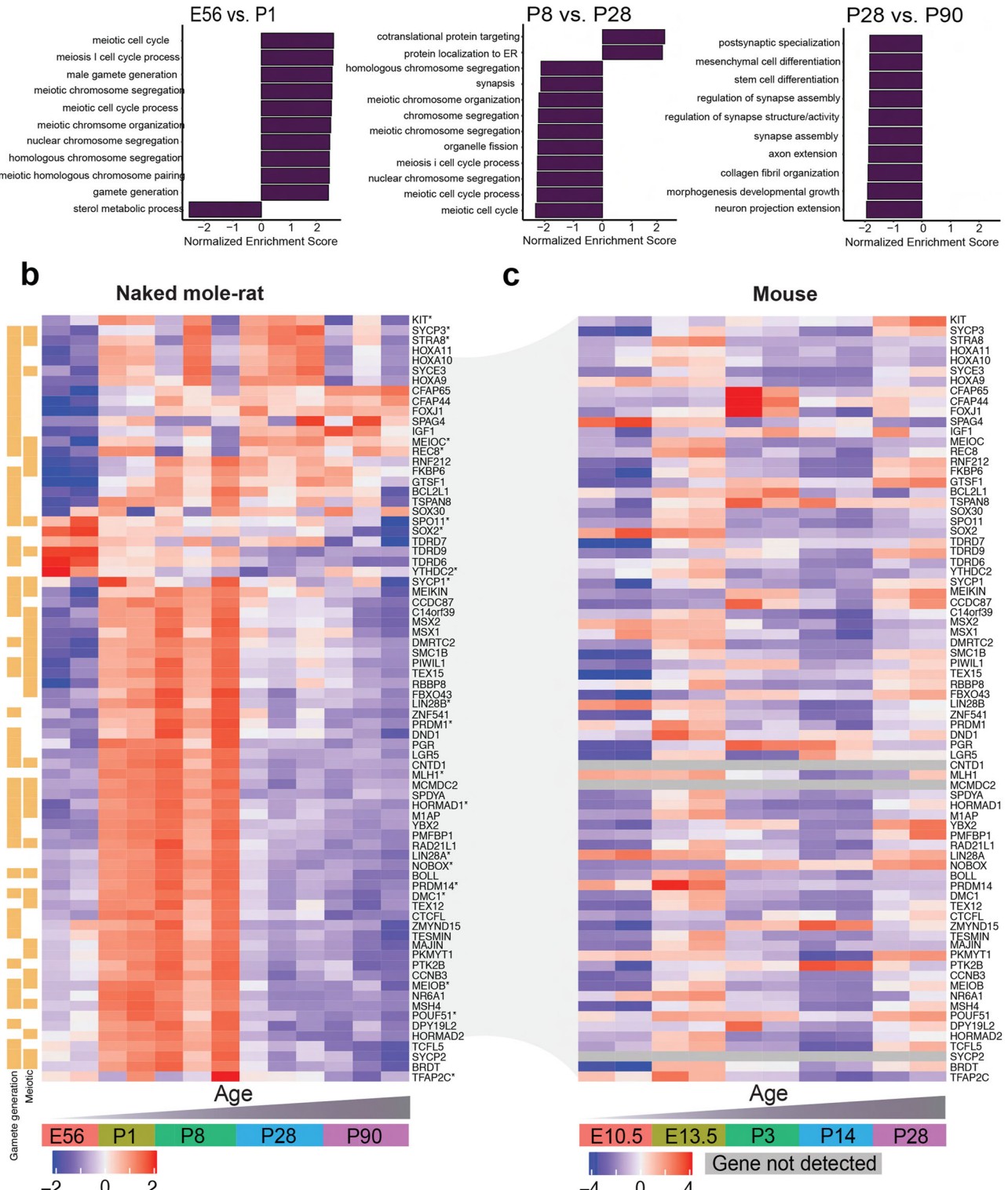

**Fig. 5 | Comparing transcriptomic profiles of ovarian development in naked mole-rats and mice. a** GSEA was performed for the ranked gene list (−1 * log2 fold-change * adjusted *p* value) of each sequential pairwise comparison using the c5 biological processes gene set. Bar plots of the normalized enrichment score show up to 10 positively- and 10 negatively-enriched pathways per comparison (ranked by normalized enrichment score; threshold: adjusted *p* value <0.05). **b** A heatmap of gene expression (counts per million) from the RNAseq data in the naked mole-rat ovary. The gene list includes the intersection of genes quantified by qPCR (labeled with asterisk) and genes that enriched for meiotic pathways from the E56 vs. PND1 gene list. Gene expression is scaled, and genes clustered by unsupervised hierarchical clustering. **c** A heatmap of the gene expression (counts per million) in the mouse ovary. Source data are provided as a Source Data File.

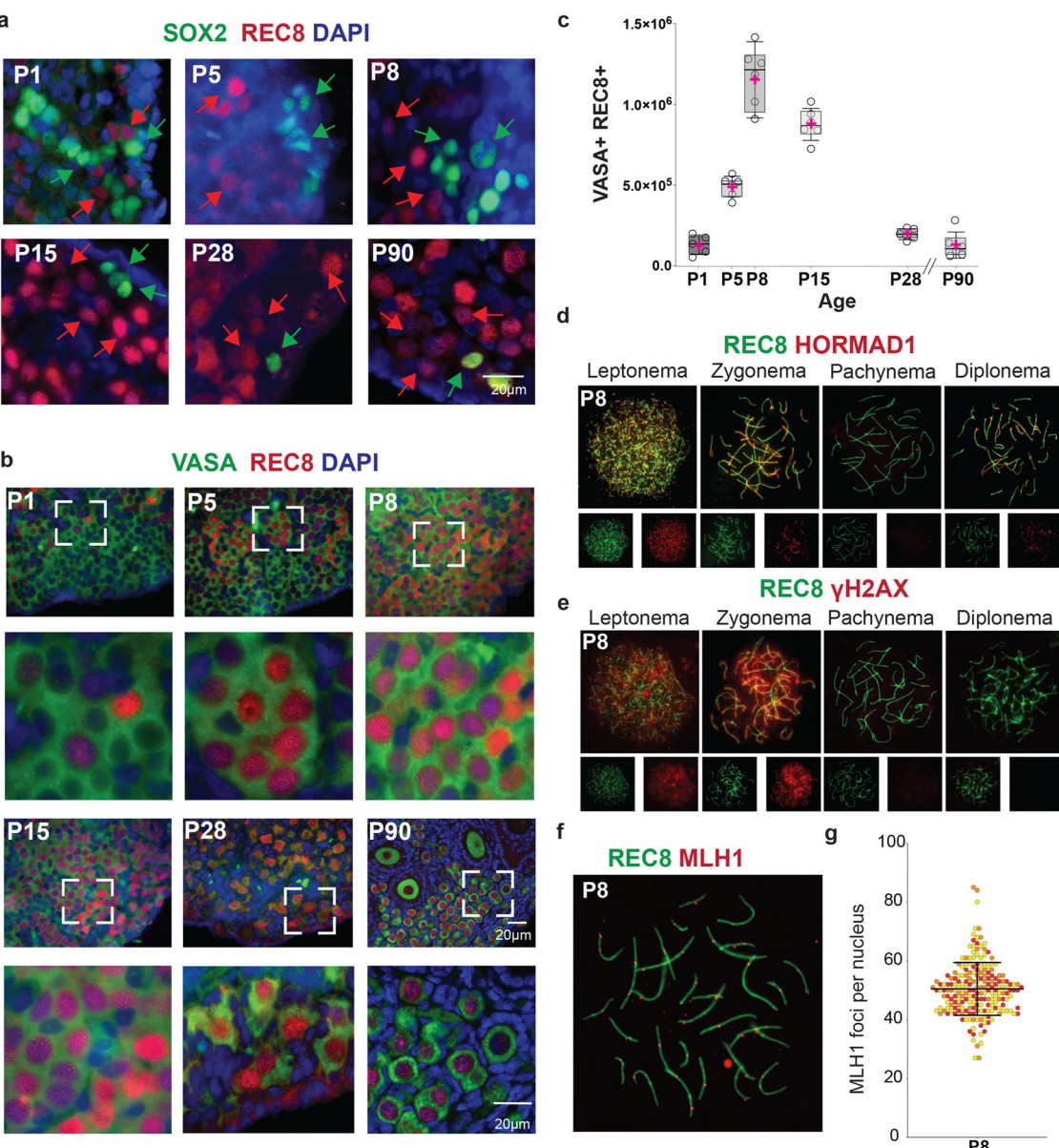

**Fig. 6 | Postnatal meiosis in the naked mole-rat (NMR) ovary. a** NMR ovary at P1, P5, P8, P15, P28, and P90, showing co-existence of cells positive for the pluripotency marker SOX2 (green) and the meiosis-specific cohesin REC8 (red). DAPI is shown in blue. Green arrows indicate representative SOX2 positive cells and red arrows indicate representative REC8 positive cells. **b** Immunofluorescence for germ cell marker VASA (green), meiotic cohesin REC8 (red), and DAPI (blue), showing cells in meiotic prophase I on tissue sections of NMR ovaries. **c** Quantification of REC8/VASA positive cells on ovaries from NMR (P1 ($n = 5$), P5, P8, P15, P28, and P90 ($n = 6$ each group)). In all the graphs, the box edges represent 25th and 75th percentiles, the horizontal line inside the box represents the median, and the pink '+' represents the mean. One-way-ANOVA, adjusting for multiple comparisons using the

Bonferroni method, all pairwise comparisons were statistically significant at $p < 0.0001$, except for age P5 vs P28 ($p = 0.16$), P5 vs P90 ($p = 0.06$), P8 vs P15, P28 vs P90, and P28 vs P1 ($p > 0.99$ for all). **d** Indirect analysis of synapsis performed by the presence of REC8 (green) and HORMAD1 (red). Presence of HORMAD1 protein indicates places where the homologous chromosomes are asynapsed. **e** Double strand break formation identified by the presence of the meiotic cohesin REC8 (green) and the marker of DNA damage γH2AX (red) and from leptonema to diplonema. **f** Immunolocalization of REC8 (green) and MLH1 (red) on chromosome spreads from oocytes at P8. **g** Quantification of MLH1 foci numbers per nucleus from P8 ($n = 300$). Each color on the graph refers to the counts from a single animal. Source data are provided as a Source Data File.

synapsis was evaluated using antibodies against REC8 and HORMAD1, the latter protein associating only with unsynapsed regions in prophase I oocytes[56,57] (Fig. 6d). During meiotic prophase I, germ cells undergo programmed formation of DSBs, which enable homologous chromosomes to synapse and crossover. Meiotic recombination is a highly regulated process, and DSBs are generated throughout the genome in an exquisitely controlled and specific fashion[50,52]. We evaluated the DSB occurrence using antibodies against γH2AX and REC8 (Fig. 6e) and found no difference between P8 and P28 ovaries. Because oocytes are very prone to meiotic defects, we evaluated the

percentages of oocytes with hallmarks of abnormal meiotic progression, such as persistence of DSBs (γH2AX) or asynapsed chromosomes (HORMAD1). Twenty percent of the NMR oocytes showed abnormal synapsis or persistent γH2AX signal at P8 and P28, with no differences between the ages (Supplementary Fig. 8a-d). Meiotic recombination results in the formation of crossovers[50], which we evaluated using the crossover marker, MLH1[58,59]. The number of MLH1 foci at P8 ($50.47 \pm 8.99$), indicating that NMR oocytes at pachynema have at least one MLH1 focus for each pair of its 30 homologues (Fig. 6f, g). Overall, in contrast to mouse and human ovarian development[1,42], our results

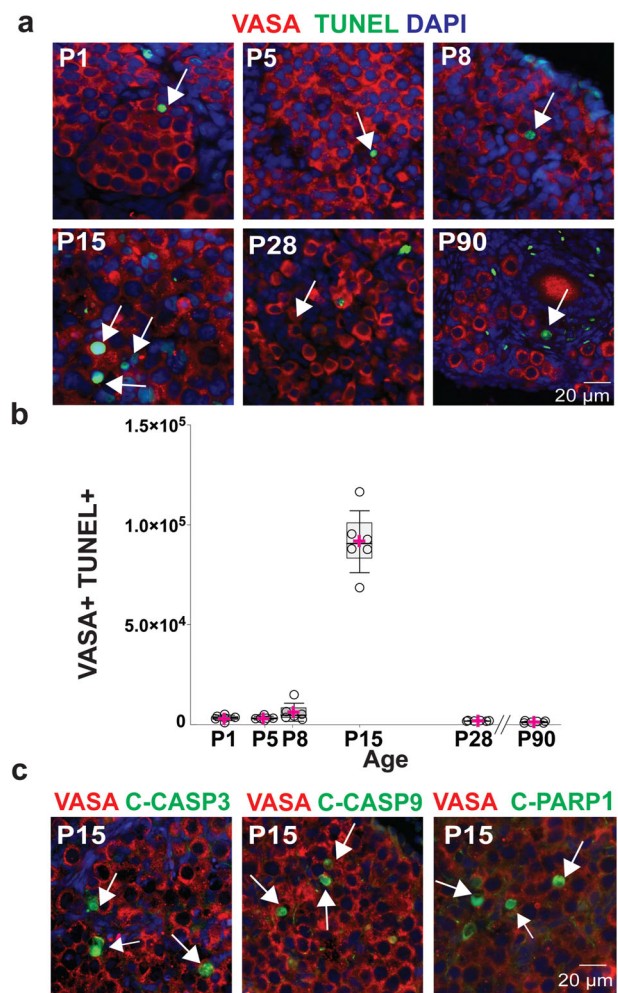

**Fig. 7 | Postnatal germ cell loss in the NMR ovary. a** Naked mole-rat ovaries at P1, P5, P8, P15, P28 and P90, showing the presence of apoptotic germ cells (VASA, red) using TUNEL (green). **b** Quantification of VASA/TUNEL cells at different stages of development (P1, P5, P8, P15, P28, P90; n = 6 per group). Box plot of TUNEL by age, the box edges represent 25th and 75th percentiles, the horizontal line inside the box represents the median, and the pink '+' represents the mean. One-way-ANOVA, adjusting for multiple comparison using Bonferroni method, only pairwise comparisons that were statistically significant at p < 0.0001 were between age 15 an all-other age. **c** P15 ovaries stained with VASA (red), DAPI (blue) and markers of apoptosis (red): TUNEL, cleaved-caspase-3 (C-CASP3), cleaved-caspase-9 (C-CAS9) and cleaved-poly (ADP-ribose) polymerase-1 (PARP1) (C-PARP1). White arrows indicate representative apoptotic cells. Source data are provided as a Source Data File.

demonstrate that PGCs and meiotic prophase I oocytes co-exist in the postnatal NMR ovary.

**Asynchronous follicle formation in the NMR ovary**

In mouse, some primordial follicles activate and start to grow soon after they form[28], and a key factor in the transition from primordial to primary follicle is NOBOX[60]. Our analysis by RNAseq and RT-qPCR showed that *Nobox* transcription starts at P8 in NMRs (Fig. 5b, c, and Supplementary Fig. 6); however, the first clear NOBOX signal by IF was observed at P15 (Supplementary Fig. 9). In mice, the transition from germ cell nest to primordial follicle formation is associated with a substantial wave of germ cell atresia/apoptosis[61]. Based on our observation in NMR that primordial follicles first form around P8 and that germ cell numbers markedly decline between P8 and P15, we expected to find the most pronounced evidence of germ cell atresia/apoptosis at P15. To interrogate NMR ovaries for germ cell atresia/

apoptosis, we co-stained P1 to P90 ovarian sections with VASA and four different apoptosis markers: TUNEL, cleaved-caspase-3 (C-CASP3), cleaved-caspase-9 (C-CASP9) and cleaved-Poly [ADP-ribose] polymerase-1 (C-PARP1) (Fig. 7a–c). Relatively few apoptotic cells were observed at P1, 5, and 8; however, we observed a marked increase in germ cell atresia/apoptosis at P15 (Fig. 7b), coincident with the significant rise in NOBOX expression. Germ cell atresia returned to low levels at P28 and P90 (Fig. 7b). Thus, these results suggest that, as in other mammals, there are processes of apoptosis/atresia during nest breakdown/primordial follicle formation and during the initial transition from primordial to primary follicles in NMRs.

**PGCs mitotic expansion in reproductively activated females**

NMRs have a very well-defined colony structure, with just one female breeder (queen) that keeps other adult females reproductively suppressed, indicating that almost all the females in a NMR colony will never be reproductively mature[16]. Our results in ovaries from P1 to P90 reveal that NMRs have a very large and asynchronous germ cell population, which includes cells positive for PGC markers, cells in meiotic prophase I, and different stages of folliculogenesis. We asked whether this phenotype is specific to early postnatal ovaries, or applies more broadly to subordinate adults and/or breeders? To answer this question, we analyzed NMR ovaries at 6 months (P6MO), which is the youngest age at which subordinate females could theoretically go through puberty if removed from the colony (i.e., released from reproductive suppression)[62]. We also analyzed 3-year-old subordinate adults (SUB 3 yr) and 3-year-old reproductively activated adults (Ex-SUB 3 yr). Both subordinates and reproductively activated females were obtained from the same colony. Following a protocol previously described[63], we reproductively activated females by housing them with a male for 4 weeks[63]. We counted the number of germ cells in histological sections, following the same approach that we performed with the ovaries from P1 to P90. We first analyzed the germ cell populations using VASA (Fig. 8a). Our results show a trend toward increased germ cells in P6MO females as compared to SUB 3 yr and Ex-SUB 3 yr females, although the values were not statistically different (p = 0.087; Fig. 8a, b). We found no significant difference in the number of germ cells in the SUB 3 yr and Ex-SUB 3 yr females. We also identified VASA+ SOX2+ cells in all three groups (Fig. 8c), and the number of cells observed on Ex-SUB 3 yr ovaries was statistically greater than in P6MO and SUB 3 yr ovaries (Fig. 8d). Next, we performed IF analyses for Ki67, pHH3, and VASA (Fig. 8e) and quantified VASA+ Ki67+ cells and VASA+ pHH3+ cells. Whereas these cell types were found in relatively low numbers in P6MO and SUB 3 yr ovaries, Ex-SUB 3 yr ovaries had significantly more cells that co-labeled for the markers of germ cell proliferation (Fig. 8f, g). We also stained for SOX2 and Ki67 cells in every 20th section, but we did not find SOX+ Ki67+ cells in the P6MO or SUB-3yr ovaries, and only a few co-labeled cells were found in Ex-SUB 3 yr ovaries (Fig. 8e). To test if VASA+ SOX2+ cells can proliferate in vitro, we performed cultures of both SUB 3 yr and Ex-SUB 3 yr ovary cortical fragments. We followed the same approach that we used for early postnatal ovaries. We began the cultures of SUB 3 yr and Ex-SUB 3 yr on the same day and under the same conditions, but the SUB 3 yr only resulted in fibroblasts from the cultured tissue. Conversely, Ex-SUB 3 yr ovary pieces extruded round cells with a large nucleolus from the main tissue starting at day 5 (D5), and small, round-cell clusters formed by D30 (Fig. 8h), which is similar to what we observed in the cultures from young ovaries. To confirm the lineage of these cells and their mitotic activity in vitro, we treated them with EdU for 48 h before fixation at D30 and D45. Fixed cultures were stained for VASA, SOX2, DAPI, and EdU, which revealed a population of VASA+ SOX2+ cells that were EdU+. This finding confirms the cells' germ cell identity and demonstrates their capability to divide in vitro (Fig. 8i).

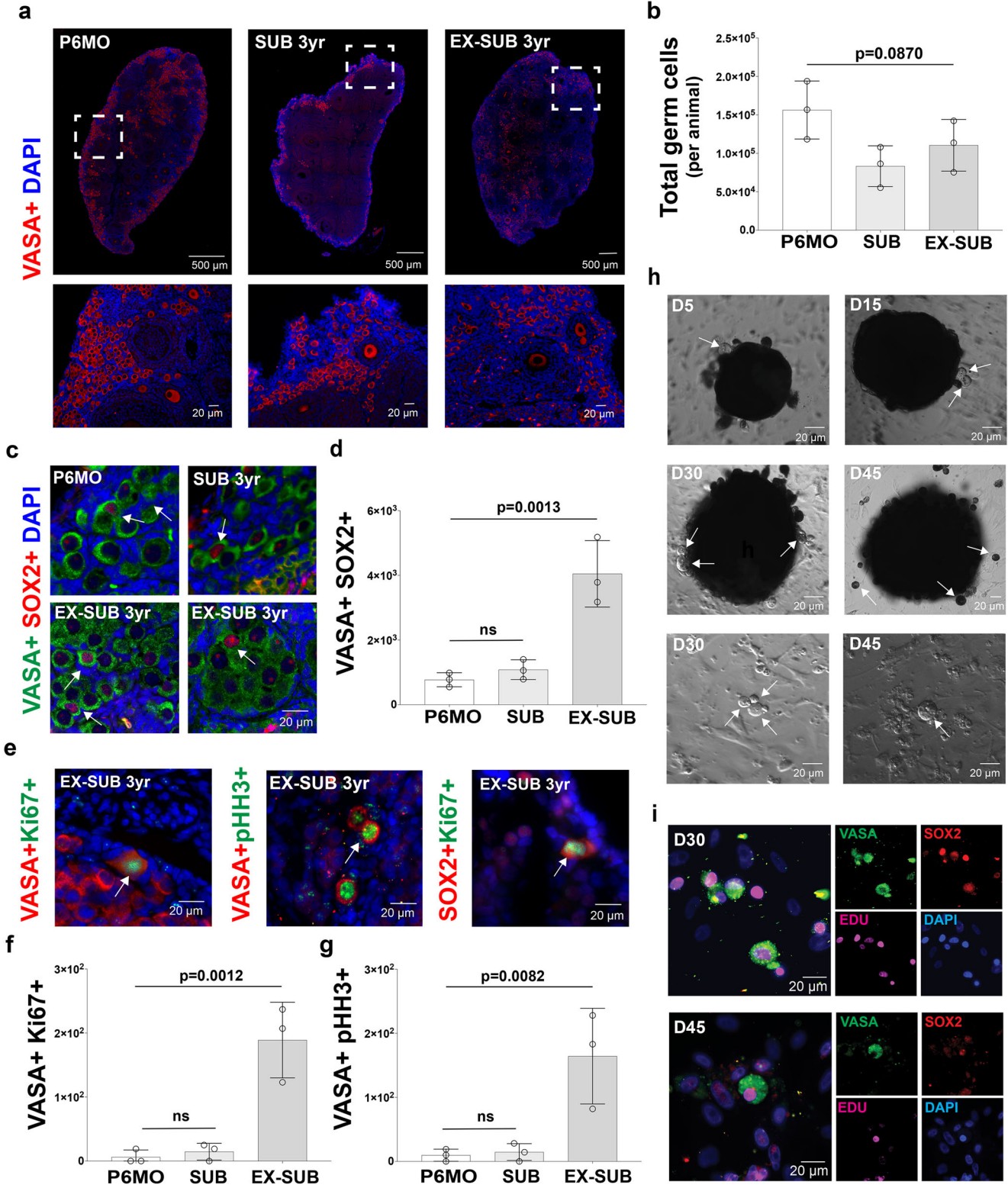

### Initiation of meiotic program in adult NMR ovaries

The presence of VASA+ SOX2+, VASA+ Ki67 and VASA+ pHH3 cells in P6MO, SUB 3 yr, and Ex-SUB 3 yr does not mean that the corresponding cells are able to undergo meiosis. Therefore, to test whether oocytes that initiate meiotic prophase I are present in ovaries from P6MO, SUB 3 yr and Ex-SUB 3 yr females, we stained histology sections to detect cells positive for VASA+ STRA8+, VASA+ γH2AX+, and VASA+ HORMAD1+ (HORMAD1 positive controls using P8, P28 and P90 are included on Supplementary Fig. 10). Our results show the presence of VASA+ STRA8+ cells in the three groups without any statistical differences (Fig. 9a, b). However, the numbers of cells positive for VASA+ γH2AX+ and for VASA+ HORMAD1+ were significantly different among Ex-SUB 3 yr and both P6MO and SUB 3 yr (Fig. 9c–f). No differences were observed between P6MO and SUB 3 yr females. These results suggest that a small proportion of cells can enter into meiosis in ovaries from P6MO and SUB 3 yr. Perhaps most interesting is that

**Fig. 8 | Six-month-old, subordinate adult, and ex-subordinate adult ovaries from naked mole-rats (NMR) have a large ovarian reserve with actively dividing germ cells. a** Histological sections of 6-month-old (P6MO), 3 yr old subordinate (SUB-3yr; reproductively suppressed), and 3 yr old ex-subordinate (EX-SUB 3 yr; reproductively activated) ovaries stained with antibody against VASA (red) and counterstained with DAPI (blue). **b** Quantification of the total germ cell number per NMR detected with VASA at P6MO, SUB-3yr, and EX-SUB 3 yr ($n = 3$ per group). One-way ANOVA shows no statistically significant difference between groups; $p = 0.870$. **c** NMR ovary sections stained with VASA (green), the pluripotency marker SOX2 (red), and DAPI (blue). White arrows indicate representative SOX2-positive cells. **d** Quantification of SOX2 positive cells at P6MO, SUB-3yr and EX-SUB 3 yr ($n = 3$ per group). One-way ANOVA, adjusting for multiple comparisons using the Bonferroni method, reveals pairwise comparisons with EX-SUB were statistically significant at $p < 0.0013$. No differences were observed between P6MO and SUB-3yr ($p = 0.7792$). **e** In vivo postnatal cell expansion in adult NMR ovaries analyzed using immunofluorescence detection of VASA (red), and marker of mitotic cell division Ki67 (green); VASA (red) and the pHH3 (green), and SOX2 (red) and Ki67 (green). White arrows indicate cells that are positive for both markers. **f** Quantification of VASA+ Ki67+ cells at P6MO, SUB-3yr and EX-SUB 3 yr ($n = 3$ per group). One-way ANOVA, adjusting for multiple comparisons using the Bonferroni method, pairwise comparisons with EX-SUB were statistically significant at $p < 0.0012$. No differences were observed between P6MO and SUB- 3 yr ($p = 0.0775$). **g** Quantification of VASA + pHH3+ cells at P6MO, SUB-3yr and EX-SUB 3 yr ($n = 3$ per group). One-way ANOVA, adjusting for multiple comparisons using the Bonferroni method, pairwise comparisons with EX-SUB were statistically significant at $p < 0.0082$. No differences were observed between P6MO and SUB-3yr ($p = 0.9895$). **h** In vitro expansion of germ cells from EX-SUB 3 yr old ovary at day D5, D15, D30, and D45 of culture. White arrows show germ cells. **i** Immunodetection of VASA (green), the pluripotency marker SOX2 (red), incorporation of ethynyl-labeled-deoxyuridine (EdU) (magenta), and DAPI (blue) in cultured cells from EX-SUB 3 yr old ovary. Source data are provided as a Source Data File.

reproductive activation of 3-year-old NMRs induces a statistically significant increase in the number of germ cells that are mitotically active and that initiate meiotic prophase I. Collectively, these results show that ovaries from reproductively active NMRs have a population of germ cells that are able to divide in vivo and in vitro and trigger meiotic prophase I, suggesting they could be capable of contributing to the maintenance of the NMR ovarian reserve. However, more studies are warranted to determine whether complete meiotic prophase progression and follicle formation occurs in adult NMRs.

## Discussion

In summary, we have found that germ cell and ovarian development in NMRs and in humans have much in common, most notably that germ cell development is highly asynchronous. This asynchrony manifests as PGCs co-existing with oogenesis, meiosis initiation, nest formation and breakdown, and follicle formation (Fig. 9g). However, in stark contrast to mouse and human, the entire process of oogenesis occurs postnatally in NMRs. We previously reported the presence of germ cell nests in 6-month-old and adult NMR ovaries[24]. Based on our observations that germ cell nests co-exist with primordial, primary, and secondary follicles in the ovaries from SUB 3 yr, Ex-SUB 3 yr, SUB 6 yr, 6 yr queen, and a SUB 10 yr (Fig. 9h, i), we suggest that oogenesis could persist for many years in this species. This hypothesis is at least partially supported by the mitotic activity of the VASA+ SOX2+ cells observed in vivo and in vitro, the meiotic prophase entry which we demonstrated in early postnatal and P6MO, SUB 3 yr, and, Ex-SUB 3 yr ovaries. However, the mechanisms by which these cells trigger the mitotic program and how SOX2 modulates other transcription factors remain to be investigated. Other important points to address in the future are, what are the differences in the ovarian niche between subordinate and reproductively activated females, and what other factors in the hypothalamic-pituitary-gonadal axis control the mitotic capacity of the germ cells in the NMR ovary?

Defects in human oocyte meiosis, and specifically during meiotic prophase I, are associated with the age-related high rate of chromosomal mis-segregation, including trisomies and monosomies, which lead to higher risks of miscarriage and developmental defects. Although aneuploidy is likely to be multifactorial, whole-chromosome nondisjunction events are preferentially associated with increased aneuploidy in young girls[64] and oocytes from older women may be predisposed to be becoming aneuploid as a consequence of an age-long decline in the cohesive ties holding chromosomes together[65]. Studying these factors in mouse and human oocytes has been limited by the fact that all these processes occur *in utero*. However, a model such as the NMR, with its pool of germ cells that initiate meiosis postnatally and in adulthood, will allow investigators to study the mechanisms of aging-associated cohesion weakening and the establishment of vulnerable crossover formations. Our preliminary data suggest that with development there is a reduction of MLH1 foci in oocytes that entered meiosis later during development (P8 vs P28), however more studies need to be done to establish the mechanism and draw conclusions. However, all these characteristics are well suited for using the NMR as a model for germ cell development and reproductive aging. Indeed, we propose the large ovarian reserve and potential for postnatal oogenesis that we report is closely intertwined with the adaptions for remarkable longevity and eusociality in this species. Naked mole-rats have the largest reproductive skew among mammals[66]; while the vast majority of adult females will never breed, all need to be ready should the opportunity arise. The impact on direct reproductive fitness is substantial for those females that attain and maintain queen status. Queens can produce 1–5 litters per year, averaging 13 pups per litter[67], which means a queen could conceivably produce thousands of offspring over her extreme reproductive lifespan. Thus, the evolutionary pressures on ovarian development and function in this species are particularly striking.

Overall, our results demonstrate that NMRs establish and maintain an exceptionally large ovarian reserve relative to their small body size, which supports their protracted fertility, and postnatal oogenesis is one of the extraordinary adaptations to this species' remarkable longevity.

## Methods

### Animals and housing

All experimental procedures followed federal and institutional guidelines and were approved by the Magee-Womens Research Institute, University of Pittsburgh (IACUC protocol #20117234), and University of Toronto Mississauga Animal Care and Use Committees (IACUC protocol # 20011632). Naked mole-rat (NMR) colonies were housed in polycarbonate cages of three sizes (large: 65 cm $L$ × 45 cm $W$ × 23 cm $H$; medium: 46 cm $L$ × 24 cm $W$ × 15 cm $H$; small: 30 cm $L$ × 18 cm $W$ × 13 cm $H$) connected by tubes (25 cm L × 18 cm $D$) and lined with corn cob bedding. For reproductive activation, opposite-sex pairs were housed in a single, medium polycarbonate cage. Naked mole-rats were kept on a 12:12 light/dark cycle at 28–30 °C and fed ad libitum with a diet consisting of sweet potato and wet 19% protein mash (Harlan Laboratories Inc.).

### Ovarian transport and dissection

All ovaries used for prophase spreads were collected in phosphate-buffered saline (PBS) at the Holmes laboratory (Toronto) and shipped overnight to the Brieño-Enríquez laboratory (Pittsburgh) at 4 °C. In the laboratory, under a laminar flux hood the ovaries were washed three times with PBS after which oviducts and connective tissue were dissected away under microscopic observation. Ovaries used for histology and immunofluorescence were dissected, fixed with 10% buffered formalin overnight and transferred to 70% ethanol, shipped to the

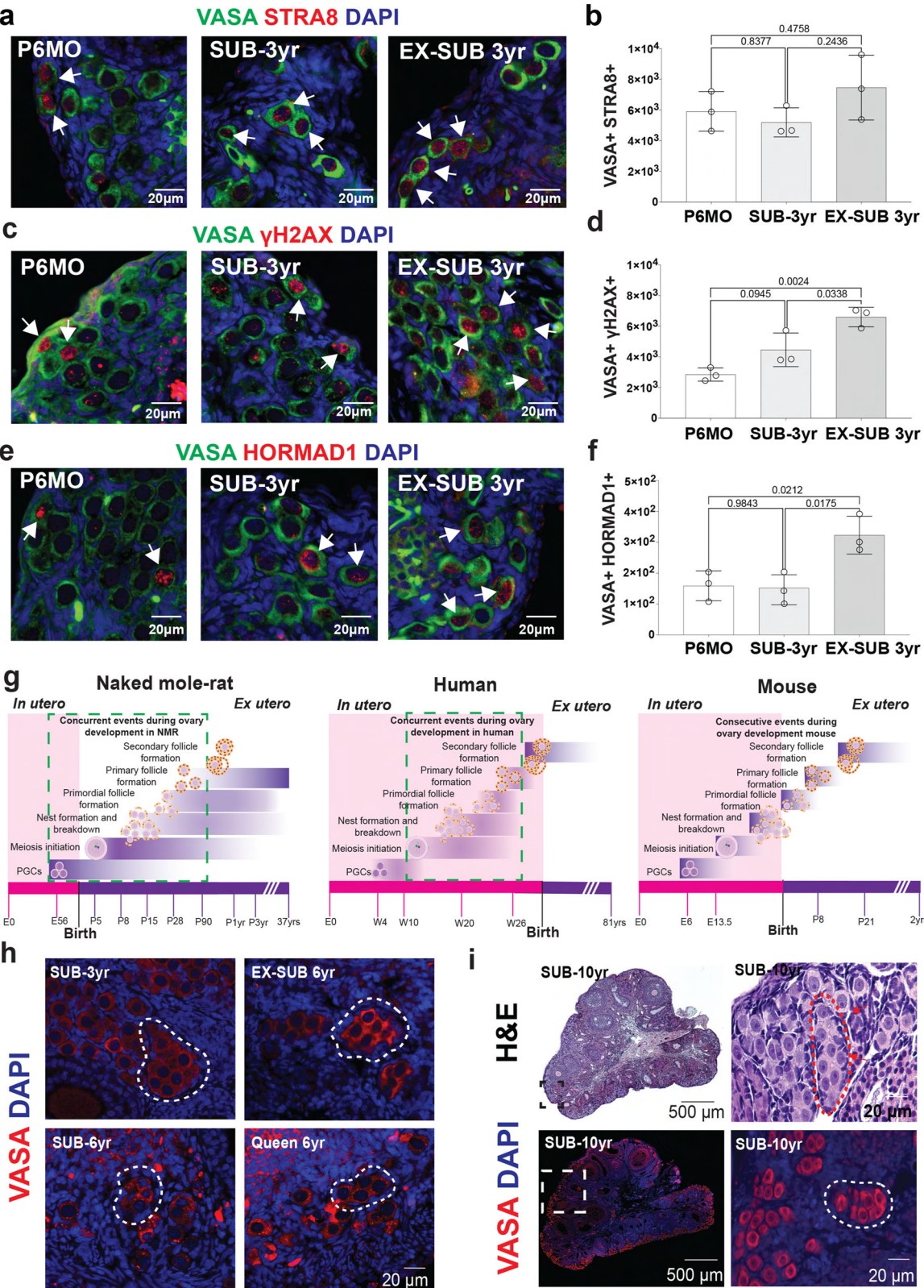

**g** Naked mole-rat, Human, Mouse — timelines of ovary development (PGCs, Meiosis initiation, Nest formation and breakdown, Primordial follicle formation, Primary follicle formation, Secondary follicle formation; In utero / Ex utero).

laboratory, washed three times in 70% ethanol, embedded in paraffin and serially sectioned at 6 μm.

## Oocyte spread preparation

To analyze meiotic progression, we cut a quarter of the ovary and placed it into 40 μl of hypotonic solution (0.02 M sucrose). Ovaries were minced with a scalpel into a homogeneous solution and resuspended in an additional 40 μl (0.02 M) sucrose. A 50 μl drop of paraformaldehyde 1% (Triton-X 0.1% pH 9.2) was added to an 8-well slide followed by the addition of 10 μl of the minced ovary solution. The slides were placed in a humidified chamber at room temperature (RT) overnight. After 24 hours at RT, the slides were washed four times with 1% Photo-Flo (Kodak, Paris, France) and air-dried. Slides were stored at −80 °C until analyzed.

**Fig. 9 | Initiation of meiotic prophase I in adult NMRs, models of ovary development in NMR, human, and mouse, and persistence of germ cell nests in NMR adult ovaries. a** Histological sections of P6MO, SUB-3yr, and EX-SUB 3 yr ovaries stained with antibody against VASA (green), STRA8 (red) and counterstained with DAPI (blue). **b** Quantification of the total germ cell number per NMR detected with VASA+ STRA8+ at P6MO, SUB-3yr, and EX-SUB 3 yr ($n = 3$ per group). One-way ANOVA shows no statistically significant. **c** NMR ovarian sections stained with VASA (green), γH2AX (red), and DAPI (blue). **d** Quantification of VASA+ γH2AX+ positive cells at P6MO, SUB-3yr and EX-SUB 3 yr ($n = 3$). One-way ANOVA, adjusting for multiple comparisons using the Bonferroni method, reveals pairwise comparisons with EX-SUB 3 yr were statistically significant at $p < 0.0024$. No differences were observed between P6MO and SUB-3yr. **e** NMR ovarian sections stained with VASA (green), HORMAD1 (red), and DAPI (blue). **f** Quantification of VASA+ HORMAD1+ cells at P6MO, SUB-3yr and EX-SUB 3 yr ($n = 3$). One-way ANOVA, adjusting for multiple comparisons using the Bonferroni method, pairwise comparisons with EX-

SUB 3 yr were statistically significant at $p < 0.0212$. No differences were observed between P6MO and SUB-3yr. White arrows indicate representative positive cells. **g** Model of NMR ovary development from E56 to adulthood. Model of human ovary development from embryonic week 4 to adulthood. Green boxes represent processes that occur simultaneously. Model of mouse ovary development, showing the consecutive events during ovary development. Pink bar at the bottom indicates processes that occur in utero and purple bar those that occur after birth. **h** Immunofluorescence with VASA (red) and DAPI (blue) showing germ cell nests (white dashed lines) in ovaries from SUB-3yr-old, EX-SUB-3yr-old, SUB-6yr-old and breeding queen 6yr-old. **i** Upper images: Hematoxylin-eosin staining of 10-year-old NMR ovary showing persistence of germ cell nests. Red arrows point to a germ cell nest. Lower images: Immunofluorescence with VASA (red) and DAPI (blue) showing a large population of germ cells. White dashed square within the left image shown at a higher magnification on the right, white dashed line shows a germ cell nest. Source data are provided as a Source Data File.

Immunofluorescence was performed as described elsewhere[43] with minor modifications. Oocytes were permeabilized using 0.1% Triton-X (Sigma) for 30 min at RT. After permeabilization oocytes were incubated in blocking solution (PBS, 2% bovine serum albumin, 0.2% goat serum and 0.05% Tween 20, 0.1% Triton-X) for 1 h at RT. Primary antibodies were diluted on the blocking solution (full list of antibodies on Supplementary Data 17) and incubated overnight at 4 °C in a humidified chamber. After washing off unbound antibodies with PBS 1% +Photoflo 1%, detection was performed with goat anti-rabbit FITC, goat anti-rabbit Cy3, goat anti-mouse Cy3, and goat anti-mouse FITC (Jackson Immuno Research Laboratories, West Grove PA, USA). Secondary antibodies were incubated for 2 h at 37 °C in a humidified chamber. Unattached secondary antibodies were washed off with PBS, and the signal was fixed with 1% formaldehyde. DNA was counterstained by applying an antifade solution (Vector Laboratories, Burlingame, CA, USA) containing 0.1 μg/ml of DAPI (4′,6′-diamidino-2-phenylindole) (Sigma). Image acquisition was performed using a Zeiss AxioImager M2 microscope (Carl Zeiss AG, Oberkochen. Germany). Images were processed using Zen 2 (Carl Zeiss AG, Oberkochen. Germany).

### Immunofluorescence on paraffin-embedded tissue sections

IF was performed on paraffin-embedded sections from E56, P1, P5, P8, P15, P28, P90, P6MO, SUB 3 yr, EX-SUB 3 yr, SUB 6 yr, Queen 6 yr, and SUB 10 yr ovaries. Slides were deparaffinized and rehydrated with 3 washes of Safeclear (Fisher Scientific, Pittsburgh, PA. USA) for 10 minutes each, followed by 3 washes of each concentration in a graded series of ethanol (100%, 95%, 80%, 70%) for 5 minutes. After rinsing the slides 2 times for 5 minutes in distilled water, the slides were incubated in sodium citrate pH 6.0 during 40 min at 95 °C. Permeabilization was performed in 0.2% of Triton-X 100 in PBS for 1 h. Sections were blocked for 4 h in blocking solution (2.52 mg/ml glycine, 10% goat serum, 3% BSA, 0.2% Tween 20 in PBS) and then incubated with primary antibody (diluted in blocking solution) overnight at RT. After 2 washes of 10 min with 0.1% Tween 20 in PBS (PBS-Tw20), the slides were incubated with secondary antibodies for 2 hours at RT. The slides were rinsed twice with PBS-Tw20 for 5 min. Cell nuclei were counterstained with DAPI and mounted in Vectashield (Vector Laboratories, Burlingame, CA).

### Germ cell counting

The size of the ovarian reserve (pre-follicle germ cells and primordial follicles) was estimated in 6 mm, serially sectioned ovaries that underwent IF for VASA, REC8, SOX2, OCT4, BLIMP1, pHH3, Ki67, C-CASP3, C-CASP9, NOBOX, STRA8, γH2AX, HORMAD1, and DAPI. Digital images were downloaded to a tablet (Surface Pro, Microsoft) and a stylus was used to tick off all the REC8 and DDX4 positive cells (or the corresponding stain) in each section using the cell counter feature in Fiji (Image J, NIH). Initially, these objects were counted in

every 10th section, but the exceptionally large number of germ cells in the P1, P5, and P8 ovaries led us to consider counting every 20th section. In an analysis of preliminary data for all five ages for which counts from every 10th and every 20th section were compared, we found an average difference of only 3.1% after every 20th section data were multiplied by two. Thereafter, the counting of germ cells up to the primordial follicle stage was completed in every 20th section. From these raw counts, we followed the method of Bristol-Gould et al.[19] to calculate the size of the ovarian reserve for each animal, whereby the total number of germ cells counted in every 20th section was divided by the number of sections counted. The germ cell number per section was multiplied by the total number of sections, which was then divided by a correction factor based on the average diameter of the objects counted. This correction factor accounts for the fact that the diameter of the objects means they could have been counted in more than one consecutive 6-mm-section. A correction factor of 1.5 was applied to P1, P5, and P8 ovaries, and correction factors of 2.0 and 2.5 were applied to P28 and P90 ovaries, respectively (for total counts see Supplementary Data 19). For P6MO and adult ovaries, we followed the same protocol, and we applied a correction factor of 2.5.

### RNA isolation

Total RNA was purified using Trizol (Thermo Fisher) according to the commercial protocol with the following changes: after the first phase separation, an additional chloroform extraction step of the aqueous layer in Phase-lock Gel heavy tubes (Quanta Biosciences); addition of 2 μl Glyco-blue (Thermo Fisher) immediately prior to isopropanol precipitation; precipitation at 4 °C for 30 minutes; two washes of the RNA pellet with 75% ethanol. RNA sample quality was confirmed by spectrophotometry (Nanodrop) to determine concentration and chemical purity (A260/230 and A260/280 ratios) and with a Fragment Analyzer (Advanced Analytical) to determine RNA integrity. RNA yields ranged from ~1 μg for E56 samples up to 25 μg for samples from older timepoints. RQN scores ranged from 2–6, however NMR shows a break in 28 S rRNA[68], generating a quality that is low for other species but is the standard for NMR. Three different ovaries were analyzed at each time point, however, we excluded E56-13 and P1-2 as these samples suggested a non-ovarian gene expression pattern (all RNAseq data is available in GEO).

### RNAseq library preparation

Ribosomal RNA was subtracted by hybridization from total RNA samples using the RiboZero Magnetic Gold H/M/R Kit (Illumina) using 100 ng total RNA and a modified protocol scaled for lower input (90 μl magnetic bead stock, and 2 μl removal solution in a 20 μl reaction). The rRNA-depleted RNA samples were then precipitated with ethanol. TruSeq-barcoded RNAseq libraries were generated with the NEBNext Ultra II Directional RNA Library Prep Kit (New England Biolabs). Each

library was quantified by Qubit (dsDNA HS kit; Thermo Fisher) with yields ranging from 100–600 ng per library.

## Quality control and processing of RNA sequencing data

Illumina Sequencing and Analysis: Libraries were pooled and sequenced on a NextSeq500 instrument (Illumina). On average, 27 M single-end 75 bp reads (minimum 20 M) were generated per library. Quality control information can be found in Supplementary Data 6 and Supplementary Fig. 4. Read quality of the raw library was assessed using FastQC (v0.11.8) (www.bioinformatics.babraham.ac.uk/projects/fastqc/ before trimming). Trimming of adapters was performed using cutadapt (v1.8)[69] (parameters: -m 50 -q 20 -a AGATCGGAAGAGCA-CACGTCTGAACTCCAGTC –match-read-wildcards). Read quality of the trimmed library was assessed using FastQC (v0.11.8). Trimmed reads were mapped to the naked mole-rat genome using RSEM (v1.2.22) and STAR aligner (v2.6.0) (https://www.r-project.org/). The NMR genome and transcriptome annotation were taken from the ENSEMBL database (release-100; ftp://ftp.ensembl.org/pub/release-100/fasta/). Following the removal of duplicate reads, reads were counted using RSEM.

## Analysis of differentially-expressed genes in the developing NMR ovary

All downstream analyses were performed in R (v4.0). Heatmaps were made using ComplexHeatmap[70]. RSEM counts were imported with tximport[71]. Libraries were filtered for mitochondrial genes and genes whose counts across samples were 3 standard deviations above the mean. Pairwise differential expression on sequential ages was performed with a likelihood ratio test in DESeq2[72] using counts per million. A full list of DEGs is available in Supplementary Data 1-4. Differentially-expressed genes (DEGs) with an adjusted $p$ value <0.05 and fold-change >log2(1.5) were considered significant (Supplementary Fig. 4)[72]. Significant DEGs from the pairwise comparisons were then plotted individually in volcano plots using ggplot2 (Fig. 4a)[73]. The top 50 upregulated genes and top 50 downregulated genes (Fig. 4b) (based on rank) per pairwise comparison were plotted in a heatmap across all ages. qPCR genes that changed significantly are labelled. Gene set enrichment for each pairwise comparison was performed using GSEA[74,75] against the c5 biological processes gene set (based on rank: −1 × log fold-change × adjusted $p$ value). The top 10 positively- and top 10 negatively enriched pathways (filtered by adjusted $p$ value; enrichment quantified using normalized enrichment score) were plotted in a bar graph (Fig. 5a; a full list can be found in the Supplementary Data 5–12). A Wald's test was used to test for differentially-expressed genes in a time-series in DESeq2[72]. DEGs with an adjusted $p$ value <0.05 and fold-change >log2(1.5) were considered significant. A full list of DEGs is available in Supplementary Data 13. DEGs were then ranked by adjusted $p$ value and the top 15 upregulated and top 15 downregulated genes (ranked by adjusted $p$ value) were plotted in a heatmap across all ages (Supplementary Fig. 5). Gene set enrichment for the time-series genes was performed using GSEA[74,75] against the c5 biological processes gene set (based on rank: −1 × log fold-change * adjusted $p$ value; a full list can be found in the Supplementary Data 14–15).

## Comparing NMR ovarian development to the mouse

The NMR was then compared to publicly available mouse (*Mus musculus*) ovarian development RNAseq data[45]. The fastq files for the mouse ovarian RNAseq data for E10.5, E13.5, P3, P14, and P28 were downloaded from https://www.ebi.ac.uk/arrayexpress/experiments/E-MTAB-6798/[11]. The data were processed using the same pipeline as the NMR RNAseq data, with the exception of using the mouse genome and transcriptome annotation (*Mus musculus*, ftp://ftp.ensembl.org/pub/release-100/fasta/; GENCODE M25). t-SNE plots for the NMR and mouse RNAseq data were then plotted using the t-SNE wrapper

function in M3C[76] (Supplementary Fig. 4d-e). Next, a gene list was compiled consisting of the 20 qPCR genes profiled in Supplementary Fig. 5 and the 65 genes that enriched for meiotic pathways in the E56 vs. PND1 comparison ("chromosome organization involved in meiotic cell cycle", "gamete generation", "homologous chromosome pairing at meiosis", "homologous chromosome segregation", "male gamete generation", "meiosis I cell cycle process", "meiotic cell cycle", "meiotic cell cycle process", "meiotic chromosome organization" and "meiotic chromosome segregation", and "nuclear chromosome segregation"). Nine genes overlapped between the two lists (*Dmc1, Hormad1, Meiob, MeioC, Rec8*, Spo11, *Stra8, Sycp1, Sycp3*), leaving a total of 76 genes. The gene expression (counts per million) for the gene list was then plotted in a heatmap in the NMR and mouse (Fig. 5b, respectively).

## Quantitative real-time PCR with reverse transcription

500 ng of total RNA was reverse transcribed using Superscript III First-Strand Synthesis System (Invitrogen Co, Carlsbad, CA, USA) and 1.65 M random hexamers/1.25 M Oligo(dt). cDNA was kept at −20 °C until used in qPCR. Gene expression was normalized to succinate dehydrogenase complex flavoprotein subunit A (*Sdha*) and mitogen activated protein kinase-1 (*Mapk1*) expression. Each reaction mixture consisted of 1 μl of cDNA, 0.5 μl of forward primer (0.2 μM), 0.5 μl of reverse primer (0.2 μM), 10 μl of Roche FastStart SYBR Green Master (Sigma-Aldrich, St. Louis, MO. USA) and 7 μl of nuclease-free water. qPCR amplification reaction was performed with specific primers (Supplementary Data 18). PCR conditions were 30 s at 95 °C, followed by 40 cycles at 95 °C for 1 s and 60 °C for 60 s. After PCR, melting curve analyses were performed to verify specificity and identity of the PCR products. All data were analyzed with the CFX-manager Bio-Rad (Bio-Rad Laboratories). All analyzed genes were performed in triplicate for each of the three ovaries at each of the six ages analyzed (E56, P1, P5, P8, 28, and P90). RT-qPCR data for gene quantification were analyzed using the relative quantification method. Gene expression was normalized against the reference genes (*Sdha* and *Mapk1*).

## Reproductive activation of adult naked mole-rats

To induce reproductive activation in non-breeding females, 3-year-old females were removed from their natal colonies and housed individually with a male in a new cage for 4 weeks[63]. After this period the animals were euthanized, and ovaries were collected as was described above.

## Ovarian cultures

Postnatal day 5 ovaries were collected in PBS at 4 °C in the Brieño-Enríquez laboratory. Ovaries from 3-year-old subordinates and 3-year-old ex-subordinate females were collected in the Holmes laboratory and shipped to the Brieño-Enríquez laboratory (Pittsburgh) at 4 °C. Within a laminar flux hood, ovaries were washed three times with phosphate-buffered saline (PBS) after which oviducts and connective tissue were dissected away under microscopic observation. After removing the connective tissue, the ovaries were incubated for 30 min in penicillin-streptomycin solution (10,000U/ml, Gibco). For P5 ovaries, they were cut in sections of approximately 1 × 1 × 1 mm and seeded on 24-well plates with D-MEM, High Glucose, GlutaMAX Supplement (Gibco BLR) supplemented with 100 UI/ml penicillin (Gibco BLR), and 100 μg/ml streptomycin (Gibco BLR) and 15% Fetal bovine serum (Sigma-Millipore) at 32 °C and 5% $CO_2$. Fifty percent of culture media was removed every 2 days and replaced with fresh media. Cultures were evaluated at 5, 10, 15, 20, 30, 60, 90 and 120 days. Adult ovaries were processed following the same protocol, the only difference was the size of the tissue that was seeded (2 × 2 × 2 mm). Culture medium exchange and analysis were performed as described for P5. Treatment with EdU was performed 48 h before the fixation. Fixation of cultures and staining was performed as was described for the whole ovary.

## Immunofluorescence of cultures

Immunofluorescence was performed as described elsewhere[43] with minor modifications. Cultures were treated with a permeation treatment using 0.1% Triton-X (Sigma) for 30 min at RT. After permeabilization cultures were incubated in blocking solution (PBS, 2% bovine serum albumin, 0.2% goat serum, and 0.05% Tween 20, 0.1% Triton-X) during 4 h at 4 °C. Primary antibodies were diluted in the blocking solution (full list of antibodies on Supplementary Data 17), and incubated for 24 h at 4 °C. After that, cultures were washed with PBS+ Photoflo 1%, during 4 h. Secondary antibodies were incubated during 2 h at RT. Unattached secondary antibodies were washed off with PBS, and the signal was fixed with 1% formaldehyde. DNA was counterstained by applying an antifade solution (Vector laboratories, Burlingame, CA, USA) containing 0.1 μg/ml of DAPI (4′,6′-diamidino-2-phenylindole) (Sigma). Detection of EdU was performed using Click-IT, following the protocol described by the manufacturer (ThermoFisher Scientific, Waltham, MA). Image acquisition was performed using a Zeiss AxioImager M2 microscope (Carl Zeiss AG, Oberkochen. Germany). Images were processed using Zen 2 (Carl Zeiss AG, Oberkochen. Germany).

## Reporting summary

Further information on research design is available in the Nature Portfolio Reporting Summary linked to this article.

## Data availability

The RNAseq data sets generated for this study are deposited in Gene Expression Omnibus (GEO) repository, under the accession number GSE139515. The mouse RNAseq data re-analyzed for this study can be found at https://www.ebi.ac.uk/arrayexpress/experiments/E-MTAB-6798/. Source data are provided with this paper.

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

## Acknowledgements

Research reported in this publication was supported by the Eunice Kennedy Shriver National Institute of Child Health & Human Development of the National Institutes of Health under Award Number R00HD090289, P50 Pilot Project award number P50HD096723 (PI K. Orwig) to M.A.B.-E., and Award Number P50HD076210 to P.E.C. NSERC CGS M and NSERC CGS D to M.F.-M., NSERC grants (RGPIN2011-402633, RGPIN 2018-04780, and RGPAS 2018-522465) to M.M.H., the Ontario Early Researcher Award to M.M.H., the W.M. Keck Foundation Award to D.J.L., the Empire State Stem Cell Fund Postdoctoral training grant through New York State Department of Health Contract # C30293GG to

M.A.B-E (PI A. Nikitin), Eden Hall Foundation and the Magee-Auxiliary Woman Scholar endowment (MARS) to M.A.B.-E. The content is solely the responsibility of the authors and does not necessarily represent the official views of the National Institutes of Health, Empire State Stem Cell Fund, Eden Hall Foundation, or the Magee-Auxiliary Woman Scholar Endowment. We want to thank Dr. David Albertini for his initial observations about NMR ovaries, Dr. Jose Luis Barbero, Dr. Attila Tóth, and Dr. Aleksandar Rajkovic for the shared antibodies. We thank Dr. Huayun Hou for helpful advice on the RNAseq analyses.

## Author contributions

M.A.B.-E., P.E.C., M.M.H., and N.J.P. conceived and supervised the project. M.F.-M., M.M.H., P.T.W., and M.A.B.-E. provided the animals, obtained, and prepared the ovaries. M.A.B.-E., A.P., P.T.W., and J.S. performed the staining, imaging, and analysis of the ovary sections. M.A.B.-E., and A.P. performed the chromosome spreads, imaging, and analysis. M.A.B.-E., M.G.D., J.L., A.M., A.P., J.S., and K.W., performed the germ cell counts, SOX2+, REC8+, OCT4+, BLIMP1+, Ki67+, PPH3+, STRA8+, γH2AX+, HORMAD1+ and, TUNEL counts. M.F.-M., M.D.W., M.A.B.-E., and J.K.G., did the RNAseq analysis and performed the qPCRs. M.A.B.-E., and P.T.W., performed the ovarian cultures (early postnatal and ex-sub). S.L. and M.A.B.-E., performed the statical analysis. M.A.B.-E., P.E.C., M.M.H., M.F.-M., M.D.W., and N.J.P. analyzed the data and wrote the manuscript with input from all the co-authors.

## Competing interests

The authors declare no competing interests.
