## [Peer Review File · Nature Communications]

Postnatal oogenesis leads to an exceptionally large ovarian reserve and protracted fertility in naked mole-ratsREVIEWER COMMENTS

Reviewer #1 (Remarks to the Author):

The manuscript by Brieno-Enriquez and colleagues describes postnatal oogenesis in naked mole-rats using a battery of immunofluorescence stainings backed by qPCR and RNA-Seq. While the study is descriptive, it marks a milestone for the research on the female reproductive tract in this cutting-edge model organism for aging and longevity research. The manuscript will be of general interest to broad readership and to experts working on reproductive aging.

The manuscript contains several issues regarding comprehensiveness of quantifications, statistics and RNA-Sequencing analysis that must be addressed.

Please find my comments below:

MAJOR points

Related to Methods

The authors have to state whether they used breeders (queens) or non-breeding females for their study.

Supplementary tables are missing in the submission.

For ALL Figures, state the number of biological replicates/animals/samples in the legends for any quantitation.

Related to Fig. 1d

The co-staining of proliferative markers at P8 is clear, however, representative micrographs are shown. A quantitation with statistics across multiple samples between P8, P1 and one of the later postnatal stages would confirm that NMR germ cells predominantly expand postnatally.

Related to Figure 2d

The representative co-stainings for OCT4 and PRDM1 suggest much more Vasa+Oct4+/Prdm1+ cells at P1 vs P8, a pattern deviating from that of SOX2, which was enumerated along with respective statistics. It would be valuable to quantify the VASA+/OCT4+ and VASA+/PRDM1+ germ cells across the authors sampled stages P1-P90 to be able to discuss embryonic transcription factor involvement in the postnatal ovarian maturation. Please include E56 counts for all co-stainings as to give a quantitative measure comparing pre- and postnatal TF dynamics.

Related to Figure 3

While the manuscript effectively uses qPCR and IHC to derive insight into the biology of naked mole rat ovarian maturation, in this respect the authors underutilize their population RNA-seq data. There are publicly available mouse and human ovarian population (=bulk) RNA-seq datasets that could be compared to the authors' own to assess alterations across species AND pre/post-natal development. Especially with regards to the oogenesis models for three species in Figure 6a-c, the authors present intriguing data for cross-species comparative biology, however, nowhere in the manuscript include murine and/or human samples in any of their stainings. At least for population RNA-Seq data this should be compensated here.

Figures 3b and 3c provide little information regarding either QC metrics of the authors sequencing datasets and downstream processing nor biologically meaningful interpretation (and are thus barely mentioned in main text). They should be replaced with more informative content, some suggestions below.

Methods state that the authors used tophat + cufflinks (man page: <http://cole-trapnell-lab.github.io/cufflinks/manual/>) to map and count transcripts, then cuffdiff (<https://www.nature.com/articles/nprot.2012.016.pdf>) for differential gene expression (DGE). While this is somewhat okay, the gold standard for a population RNA-Seq workflow is RSEM + edgeR/DESeq2 + limma. RSEM introduced expectation maximization, greatly improving read count precision, and comfortably providing TPMs as quantification output (<https://deweylab.github.io/RSEM/README.html>). TPM have superseded RPKM and should be used throughout the authors inter-species RNA-Seq analysis, see [Wagner GP, Kin K, Lynch VJ: Measurement of mRNA abundance using RNA-seq data: RPKM measure is inconsistent among samples. *Theory Biosci.* 2012; 131(4): 281–285]. CPM derived from RSEM's `expected_counts` read count output can be used to compare intraspecies timecourses etc. Limma's advantage is ability to perform multiple comparisons within a complex experiment (such as time-course/several differentiation stages/cell types) by design of the contrast matrix, check respective literature/man pages.

Alternatively, Kallisto + Sleuth can do all analysis much faster (less computationally exhaustive) with same robustness as the tools above, however, comparing multiple groups or time courses might be difficult, please check respective sources.

The minimal QC visualization (for the supplements) would be TMM/RLE-normalized log-CPMs/TPMs across ALL samples for a given analysis in ONE plot, i.e. the intraspecies comparison across time is one graph, the inter-species comparison (authors evaluate which of their time points match to public datasets) is another.

Please show a t-SNE clustering instead of PCA for Figure 3a, improving resolution contrasts for your population RNA-Seq groups E56-PND90.

Please show a heatmap with the top 50 Up-regulated features (differentially expressed genes; DEGs) per group, only label features which have qPCR confirmation data. See if downregulated features have anything interesting.

The authors state: "Lin28a, Lin28b and Tfp2c were evaluated by RNAseq and qPCR and showed a similar expression profile to Sox2, Oct4 and Blimp1..."; "The analysis RNAseq and qPCR of Stra8 expression [...]". The proposed RNA-Seq data for the mentioned genes cannot be found in the data, thus please provide an additional supplementary section to plot

ALL 15 genes for which qPCR data has been shown, with their respective RNA-Seq data in a heatmap. Explain why, if there are differences between the two methods.

Please show top 10 gradually up- and downregulated genes across the postnatal ovary maturation, starting from E56, using heatmaps. Interpret the results with respect to known gene expression alterations during ovary development in mouse and man.

Please perform GSEA (NOT simple GO, which does not take expression and significance rank into account) on MSigDB collections C5_biological_processes with all DEGs for each time course groups. Then with complete C2 collection of all DEGs from cross-species comparison.

Related to Supplementary Figure 5

The authors state: "... the first clear NOBOX signal by IF was observed at P15 (Supplementary Figure 5)." There is no panel for P15 in this Figure. Please complete the representative images and perform quantitation as in Figures 2 and 4, then state number of animals per group.

Minor points

Related to Supplementary Figure 3

Please provide quantitation and statistics for VASA+/STRA8+ as done in Figures 2 and 4.

Correct mislabeling in Figure 4 legend: "... and cleaved-caspase 1 (C-PARP1)."

Correct typo in Figure 6 reference: "...the ovaries from a 10-year-old NMR (Figure 6c-e)."

Explain why P15 has less VASA+ cells than P8 AND P15? If it is because TUNEL is highest in P15, then KI67+ cells should be enhanced in P28 vs P15 and P90?

Related to Figure 5

While the prophase I chromosome spreads are impressive, their discussion in the text is insufficient for understanding. What is leptoneuma, zygonema etc (stages of prophase I), why are they different, and how can this be quantified? gammaH2AX is not different between P8 and P28 as per Supplementary Figure 4a, but where are the error bars in Supplementary Figure 4b?

The conclusion presented for result from Figure 5c/d is unsatisfactory: Why has P28 still quite similar MLH1 foci number to P8, while the germ cells (VASA+, Figure 2d) are drastically different? And when REC8+VASA+ cells mark oocytes, and Figure 4d shows

presence of oocytes throughout all time points with a similar pattern as total germ cells (Figure 2d), what is the relevance of the Figure 5d result?

Reviewer #2 (Remarks to the Author):

Review of Brieno-Enriquez et al.

Summary

In the vast majority of mammals such as mouse, primates and humans, females are born with a reserve of follicles (oocytes and their surrounding cells) that deplete over their life span. Once this reserve is depleted, females reach reproductive senescence- in human this is known as menopause. It is therefore of interest to understand ovarian aging in different species. Ovarian aging is characterized by hormonal dysregulation, such as decreased sex steroids. A second layer of complexity is added by fertility rates, which are usually flat with female age- meaning that across a population, females reproduce until they die (or undergo menopause). Like the vast majority of mammals, naked mole rats (NMR) have a flat rate throughout their lifespan. What is different about NMR is that they live for decades and yet have high fertility rates until they reach their end of life span. In inbred mouse (in captivity), litter sizes ('fertility') decline after 6-9 months. It is therefore of interest to understand how litter sizes and fertility rates are maintained for three decades in the NMR.

Overall assessment:

In this study, the authors provide a cytological characterization of germ cell development in the naked mole rat (NMR) ovary during the postnatal stages (P1 to P90- 3 months). They follow the dynamics of total germ cells (VASA+) from three animals at P1, P5, P8, P15, P28 and P90 and conclude that the ovarian pool is established in the post-natal ovary, as opposed to the pre-natal ovary as is seen in mice and humans. They further propose that the prolonged fertility in the NMR may be due to the persistence of PGCs/OSCs into later ages. This difference in timeline and fertility makes the NMR an interesting tool for studying early oogenesis and female reproductive aging.

Major comments:

1. PGC expansion in the postnatal ovary

One of the major strengths is the meiotic analysis conducted. The conclusion that meiotic prophase is taking place in the perinatal ovary is based on strong evidence from the Cohen lab. In contrast, the conclusion that naked mole rats expand their PGCs postnally hinges on Figure 1, where the data and their statistical analyses are not clear.

In Figure 1a-c, the authors count total germ cells (VASA+) from three animals at P1, P5, P8, P15, P28 and P90. At P8, the three animals appear to have a higher number of total germ cells than at P5 and the authors conclude that there is an increase in PGCs/GCs around P8. However, the comparison of P8 with P1 does not yield a 'significant' p-value. Furthermore, there seems to be some technical issues with the ovary fixation at the critical P5- the H&E in Fig 1a doesn't look good. Is this just this specific section? Problems with fixation would also influence antibody staining of P5 in Fig. 1b, where the VASA staining at P5 looks as if the antibody did not penetrate the tissue properly.

causing an underestimate and could provide a rational explanation of the apparent low count in P5 compared to P8. More fundamentally, with only three animals per stage (P1, P5 and P8), this conclusion should be further supported- the data are not clear from these low numbers.

The assumption that VASA+ cells increase at P8 has downstream ramifications on the inference that pH3 and Ki67 markers are proliferating cells. These markers would also be present in meiotic cells, as VASA. It is therefore questionable whether further PCG expansion occurs in the postnatal ovary or whether it has all occurred in the fetal ovary.

Finally, a basic tenet of stem cells is that they should be capable of proliferating in vitro. This should be demonstrated for the PGCs/OSCs (Sox2) that the authors suggest are oogonial stem cells.

Taken together, the data presented support the conclusion that meiotic prophase takes place postnatally, but do not convincingly demonstrate that postnatal PGC/OSC expansion occurs.

2. Fertility preservation throughout life span

It is unclear how the authors are suggesting NMR preserve their fertility throughout life. In the model (Figure 6), it looks as if they suggest NMR establish their entire reserve during the first 3 month of their perinatal life. However, in the text they write that oogenesis could persist for many years (line 177). What is the authors' model?

The ambiguity might in part be due to that their major data stop at P90 and they have a single ovary from a 10-year old female presented in Figure 6, coming after their model. The staining shows that germ cell nests (VASA positive) appear to be present in a 10-year old ovary. They further use this single ovary to suggest that 10-year old NMR has 5-times more germ cells than mouse at 2 years (line 179). This last statement should either be removed or further data provided. A single data point cannot be taken as a given for an entire species (or colony). The manuscript would be much improved if the authors communicate clearly the various explanations for their observation- the VASA+ cells could be senescent, they could also be proliferating and initiating oogenesis throughout life. Again, demonstrating that these cells are capable of proliferating, a basic tenet of stem cells, should be demonstrated. At the very least further characterization and co-staining (e.g. SOX2) as well as an increased number of ovaries even from younger females (6 months, 1 year) would be needed to make firm conclusions about potential PGCs/OSCs. At the very least, one cannot conclude that NMRs maintain their fertility by continuous OSC activity throughout their life.

3. Statements in the manuscript.

The entire manuscript would benefit from removing statement that focus on how different naked mole rats are compared to other mammals and making them an example that postnatal OSC expansion exists. An example is the last sentence of the Abstract that germ cell proliferation occurs in the postnatal mammalian ovary. This extrapolation is not only inaccurate as many species do not contain OSCs.

There are several factual statements that are unclear in the Introduction and Discussion. This includes when females are fertile. Line 52-54 state no decline into their third decade of life, whereas line 60, says that they are fertile across their three decades-long lifespan. These two statements seem at odds with each other.

There are several statements that need to be changed. For example, although women have lost 90% of the follicles by 30, they are still fertile- in fact, it's the peak for fertility in women. What is meant is that women have an extensive post-reproductive lifespan (post-menopause). These are but a few examples.

4. Statistics.

There were many examples where statistical comparisons had been carried out and the p-values but not effect sizes nor power calculations shown. Neither is it clear whether p-values had been adjusted when conducting multiple comparisons. It was difficult to evaluate objectively the biological significance in several of the experiments, especially in Figure 1.

Minor comments:

Figure 1. It's difficult to appreciate that the ovaries grows with the scale bar at different sizes.

Figure 1a: the H&E staining of the P5 looks strange. Is this what you typically see at that stage? Please see my major comment.

Figure 3 was poorly utilized. Information from the RNAseq analysis was not discussed in the paper. All of the expression data discussed was from supplemental figure 3; Moving figure S3 into the main text to replace the current figure 3 might make more sense.

It would help the readability of the manuscript to tone down the differences with other rodents throughout.

There are many repetitions that should be removed. E.g. line 60 has already been explained in the beginning of the paragraph.

Please remove 'Interestingly', 'remarkable'.

Line 196: the laboratory. Which one(s)?

Line 261: Total RNA isolation. Please specify what tissue, how much, and what cell type composition this tissue would be expected to have.

Line 333: I think Attila Toth's name is misspelt (if that is who you are thanking).

Some of the figures are mentioned out of order. For example, the TUNEL assay in figure 4d and e was found in the text after all of the meiosis analysis.

There were several points where the wrong figure was listed.

There were several points in the text where figure 3 was references, but the text was discussing figure 4.

The paper would benefit from a deeper discussion on the use of this model organism in the field. The final paragraph was a nice summary of the study, but it's unclear whether much can be extrapolated.

Reviewer #3 (Remarks to the Author):

The manuscript presents a characterization of oogenesis in female naked mole rats. The main findings of the manuscript are that germ cells remain mitotically active at least for 8 days after birth, and that meiosis is initiated in germ cells only 8-28 days after birth. These properties of naked mole rat oogenesis contrast oogenesis in mice and humans, where germ cells stop mitotic divisions and complete meiotic prophase in foetal ovaries. The manuscript also reports germ cell nests in 10 years old animals, which raises the possibility that germ stem cells or mitotic germ cells persist well into adulthood in naked mole rat. However, the cell cycle status of these adult germ cells is not determined, thus it is not clear if they are mitotic or meiotic cells, and/or they are in a resting phase having past through meiotic prophase. The manuscript also finds an exceptionally large ovarian reserve in 8 days old naked mole rats, which may provide explanation for protracted fertility in this species.

The presented data provides sufficient evidence for the main conclusion that there is postnatal germ cell proliferation and meiosis initiation in female naked mole rats. This finding is interesting to the reproduction biology field and is consistent with other peculiar features of naked mole rats, which probably, co-evolved with their eusociality. However, I am not certain that the depth and the importance of the discoveries of the manuscript justify access to the wide readership of Nature Communications. The manuscript may fit better a specialized journal, such as Biology of Reproduction or Developmental Biology. This is because the main findings do not reveal a very surprising or a generalizable phenomenon. The main discovery is that the developmental timing of meiosis initiation is shifted to a slightly later age in naked mole rats as compared to humans or mouse. Nonetheless, the general pattern of naked mole rat oogenesis seems similar to oogenesis in other mammals: intense mitotic proliferation of germ cells takes place in immature animals, proliferation seem to cease or nearly cease in juveniles and germ cells go through meiotic prophase in immature animals, hence a finite oocyte pool is generated that seem to support fertility throughout mature life. The slightly delayed timing of this developmental pattern in a highly specialized animal species is not very surprising given the already reported differences between oogenesis in humans and mice, which are summarized in figure 6. Thus, while the manuscript represents a descriptive study of oogenesis in a species that is interesting for longevity and regeneration studies, it does not uncover a truly novel or surprising concept.

Beyond the above general concern, I would have several suggestions for improvements in the manuscript.

1. The presentation of the ovarian transcriptome analysis in figure 3 is not very informative as it only shows that the transcriptome changes during development, which is not a very revealing conclusion. Including this data to the main figure would be justified only if the manuscript presented more analysis of RNAseq data. For example, the figure could present changes in the expression of various marker genes for PGCs, mitotic germ cells and meiotic germ cells matching the qPCR analysis in supplementary figure 3.

2. The manuscript present data on mitotic proliferation 8 days after birth and meiotic prophase progression only from 8 or 28 days old animals. It is somewhat ambiguous in the text, but my understanding is that all immature stages were tested for mitotic proliferation, but proliferation was seen only in P8 animals. In contrast, meiotic prophase was only looked at in P8 and P28 animals. These analyses are incomplete without testing other developmental stages including 90days old or mature animals. Are germ cell proliferation and meiotic initiation take place throughout the life of naked mole rats or they are restricted to immature stages? Further, I may have missed it, but does the manuscript show histology of ovaries from a 10 year old breeder queen or a non-breeding worker in figure 6. To get an accurate picture of how oogenesis is adapted to fit the unique eusociality of naked mole rats, it would be essential to examine the oocyte pool in both adult workers and queens. Is there a difference in the oocyte pool size in adult queens and workers? Is there differential mitotic germ cell proliferation and germ cell pool replenishment in adult workers and queens? Apparent cessation of mitotic divisions in immature animals that are destined to become sexually inactive workers might not reflect the status of germ cells in adult queens. Thus, are there dividing mitotic germ cells in adult queens? Finally, are there significant numbers of germ cells that initiate meiosis in adult queens as compared to workers? These are all very important questions that are left unanswered by the manuscript, which severely limits the depth of the conclusions and the value of the manuscript.

3. Connected to point two, analysis of germ cell nests from the 10 years old adult is incomplete. It is not clear what cell cycle status germ cells have in the nests. It is also not clear if germ cell nests are present in both queens and workers or only in workers assuming that the presented image is from a worker. It would not be surprising if folliculogenesis was delayed in non-breeder workers as compared to breeding queens.

4. It would improve the manuscript if there was more discussion about the possible interplay between the eusociality and the developmental pattern of oogenesis in naked mole rats. In particular, how the life cycle of the naked mole rats permitted and/or necessitated changes in the timing and regulation of oogenesis in worker and breeder naked mole rats as compared to their rodent relatives.

REVIEWER COMMENTS

We thank the Reviewers for their positive reviews and constructive critiques. We also want to thank you for the additional time to complete the experiments needed in order to complete a comprehensive revision that addresses all of the comments made by the Associate Editor and Reviewers. This work is a collaborative effort between labs in Canada and United States, and as you know, the border between the two countries was closed during the pandemic and access to our laboratories was severely restricted. Our inability to even access the naked mole-rats (NMR), import their samples from Canada to the United States, and to perform experiments and analyses resulted in substantial delays in completing the revision of our manuscript. Thank you so much for your patience.

Reviewer #1 (Remarks to the Author):

The manuscript by Brieño-Enriquez and colleagues describes postnatal oogenesis in naked mole-rats using a battery of immunofluorescence staining backed by qPCR and RNA-Seq. While the study is descriptive, it marks a milestone for the research on the female reproductive tract in this cutting-edge model organism for aging and longevity research. The manuscript will be of general interest to broad readership and to experts working on reproductive aging. The manuscript contains several issues regarding comprehensiveness of quantifications, statistics and RNA-Sequencing analysis that must be addressed.

We want to thank the reviewer for all the comments and constructive criticisms.

Please find my comments below:

MAJOR points

Related to Methods

R1. Q1: The authors have to state whether they used breeders (queens) or non-breeding females for their study.

In the original manuscript all the data were from fetal and juvenile animals (E56-P90), and earliest age at which a female can be reproductively activated is 6 months. The pictures of the 10-yr-old ovary corresponded to a non-reproductive, subordinate female (lines 362-365). In the revised manuscript, we clearly state ovaries from P1 to P90 are from females that are too young to be reproductively activated. We now include data from P6MO, SUB-3yr and Ex-SUB-3yr females, with the latter having undergone reproductive activation. (Lines 314-326)

R1. Q2: Supplementary tables are missing in the submission.

Supplementary tables are now added to the submission

R1.Q3: For ALL Figures, state the number of biological replicates/animals/samples in the legends for any quantitation.

We added “n” numbers to all figure legends

R1. Q4: Related to Fig. 1d. The co-staining of proliferative markers at P8 is clear, however, representative micrographs are shown. A quantitation with statistics across multiple samples between P8, P1 and one of the later postnatal stages would confirm that NMR germ cells predominantly expand postnatally.

In the revised manuscript version, we included the analysis of the proliferative markers for all ages from P1 to P90 (lines-172-182). In Figure 1, we included representative images, as well as quantifications and analyses of VASA+Ki67+ across the ages. The same information has been provided for VASA+pHH3+ (Supplementary Figure 1) and Ki67+SOX2+ (Figure 2).

R1. Q5: Related to Figure 2d. The representative co-staining for OCT4 and PRDM1 suggest much more Vasa+Oct4+/Prdm1+ cells at P1 vs P8, a pattern deviating from that of SOX2, which was enumerated along with respective statistics. It would be valuable to quantify the VASA+/OCT4+ and VASA+/PRDM1+ germ cells across the authors sampled stages P1-P90 to be able to discuss embryonic transcription factor involvement in the postnatal ovarian maturation. Please include E56 counts for all co-staining as to give a quantitative measure comparing pre- and postnatal TF dynamics.

We performed co-staining, quantifications, and analyses of the different populations of cells. Our analyses include co-staining of VASA+SOX2+, VASA+OCT4+, VASA+BLIMP1+ and SOX2+/OCT4+ (lines 147-172, Figure 2). As the reviewer pointed out, initially there is a shift in the expression of the different markers, at early ages all of them are present, but with development the predominant transcription factor is SOX2. We agree that the balance of these transcription factors and the ages when they're present are key factors to consider for self-renewal and differentiation. We are actively working on the potential mechanisms that regulate them in the NMR ovary, and it is part of our ongoing research. We agree that analysis of fetal samples will provide more information about the dynamics of the TFs, however obtention of fetal samples in the NMRs is very challenging because it requires the sacrifice of a colony's only breeding female (the queen). The establishment of new queens and requires at least 2 months to initiate breeding but is unpredictable and can take as long as 3 years. Successfully breeding queens are a scarce and highly valued resource in NMR research. Additionally, we had very limited access to the NMR colonies for nearly two years due to the pandemic and we were not allowed to expand colony breeding according to University of Toronto COVID-19 regulations (to reduce workload on essential vivarium staff). However, our results on postnatal ovaries showed co-localization of SOX2+ and OCT4+ was not statistically different at P1 and P5.

R1. Q6: Related to Figure 3. While the manuscript effectively uses qPCR and IHC to derive insight into the biology of naked mole rat ovarian maturation, in this respect the authors underutilize their population RNA-seq data. There are publicly available mouse and human ovarian population (=bulk) RNA-seq datasets that could be compared to the authors' own to assess alterations across species AND pre/post-natal development. Especially with regards to the oogenesis models for three species in Figure 6a-c, the authors present intriguing data for cross-species comparative biology, however, nowhere in the manuscript include murine and/or human samples in any of their stainings. At least for population RNA-Seq data this should be compensated here.

We thank the reviewer for their comments and advice on the RNA-seq analyses. Following the advice of the reviewer (see below for details), we completely re-analyzed the RNA-seq data and performed a comparative analysis using publicly available mouse ovarian gene

expression data from Cardoso-Moreira et al., 2019 (DOI: [10.1038/s41586-019-1338-5](https://doi.org/10.1038/s41586-019-1338-5)). From this dataset we carefully selected 5 mouse ovarian development timepoints that are most comparable to the naked mole-rat timepoints presented in this study. Note that the human ovarian developmental datasets we could find were not comparable as they only included embryonic timepoints. We used the same pipeline to re-analyze the mouse ovarian data (see R1. Q7 for details).

R1. Q7: Figures 3b and 3c provide little information regarding either QC metrics of the authors sequencing datasets and downstream processing nor biologically meaningful interpretation (and are thus barely mentioned in main text). They should be replaced with more informative content, some suggestions below.

We thank the reviewer for these helpful suggestions. We believe this has strengthened our paper. These new analyses are shown in new Figures 3 and 4, Supplementary Figures 4 and 5, the Methods section (lines 496-530), and Supplementary Tables 1-16. Further description of these results is provided in the main text (see below).

R1. Q8: Methods state that the authors used tophat + cufflinks (man page: <http://cole-trapnell-lab.github.io/cufflinks/manual/>) to map and count transcripts, then cuffdiff (<https://www.nature.com/articles/nprot.2012.016.pdf>) for differential gene expression (DGE). While this is somewhat okay, the gold standard for a population RNA-Seq workflow is RSEM + edgeR/DESeq2 + limma. RSEM introduced expectation maximization, greatly improving read count precision, and comfortably providing TPMs as quantification output (<https://deweylab.github.io/RSEM/README.html>). TPM have superseded RPKM and should be used throughout the authors inter-species RNA-Seq analysis, see [Wagner GP, Kin K, Lynch VJ: Measurement of mRNA abundance using RNA-seq data: RPKM measure is inconsistent among samples. *Theory Biosci.* 2012; 131(4): 281–285]. CPM derived from RSEM's expected_counts read count output can be used to compare intraspecies timecourses etc. Limma's advantage is ability to perform multiple comparisons within a complex experiment (such as time-course/several differentiation stages/cell types) by design of the contrast matrix, check respective literature/man pages. Alternatively, Kallisto + Sleuth can do all analysis much faster (less computationally exhaustive) with same robustness as the tools above, however, comparing multiple groups or time courses might be difficult, please check respective sources.

We thank the reviewer for these valuable methodological suggestions. We have implemented these suggestions and believe they have improved the manuscript. Firstly, we completely re-analyzed our RNA-seq data making use of STAR + RSEM. In addition to analyzing differential expression between sequential timepoints (using DESeq2; Figures 3 and 4); we also explored using edgeR + limma (using an intra-species time course design) and DESeq2 (using a pairwise, sequential time-series design; Supplementary Figure 5). This analysis is now described in response to R1. Q12.

R1. Q9: The minimal QC visualization (for the supplements) would be TMM/RLE-normalized log-CPMs/TPMs across ALL samples for a given analysis in ONE plot, i.e. the intraspecies comparison across time is one graph, the inter-species comparison (authors evaluate which of their time points match to public datasets) is another. Please show a t-SNE clustering instead of PCA for Figure 3a, improving resolution contrasts for your population RNA-Seq groups E56-PND90.

We thank the reviewer for this suggestion and agree that additional QC visualization will help in the assessment of our RNA-seq experiment. QC was performed using RSEM-derived RLE-normalized log CPM. We have now included both PCA and tSNE plots for our

original NMR RNAseq data and the published mouse data along with pairwise correlation of replicates (Supplemental Figure 4). Additional RNA-seq QC metrics are now included in Supplementary Table 16.

R1. Q10: Please show a heatmap with the top 50 Up-regulated features (differentially expressed genes; DEGs) per group, only label features which have qPCR confirmation data. See if downregulated features have anything interesting.

We thank the reviewer for this helpful suggestion. We have now included the top 50 up-regulated features and top 50 down-regulated features (Figure 3b). Both heatmaps include a legend to mark genes that have qPCR confirmation data. The qPCR genes have been labelled on this figure to better connect these results with the rest of the manuscript.

R1. Q11: The authors state: “Lin28a, Lin28b and Tfap2C were evaluated by RNAseq and qPCR and showed a similar expression profile to Sox2, Oct4 and Blimp1...”; “The analysis RNAseq and qPCR of Stra8 expression [...]”. The proposed RNA-Seq data for the mentioned genes cannot be found in the data, thus please provide an additional supplementary section to plot ALL 15 genes for which qPCR data has been shown, with their respective RNA-Seq data in a heatmap. Explain why if there are differences between the two methods.

We now show the RNAseq expression for all the qPCR genes on the Figure 3b and heatmaps from both NMR and mouse Figure 4.

R1. Q12: Please show top 10 gradually up- and downregulated genes across the postnatal ovary maturation, starting from E56, using heatmaps. Interpret the results with respect to known gene expression alterations during ovary development in mouse and man. Please perform GSEA (NOT simple GO, which does not take expression and significance rank into account) on MSigDB collections C5_biological_processes with all DEGs for each time course groups. Then with complete C2 collection of all DEGs from cross-species comparison.

We thank the reviewer for these suggestions. We now show the top 15 gradually up- and downregulated genes which were identified using DESeq2’s time-series formula in a heatmap (Supplemental Figure 5), as well as perform a comparative gene expression analysis with mouse.

a) We performed GSEA (using the MSigDB collections C5_biological_processes) of the time-series analysis as well as for the sequential pairwise gene expression analyses (Figure 3). From both analyses we see evidence for changes in pathways related to meiosis and gametogenesis. In particular, we found upregulation of meiotic/gamete generation pathways in PND1 ovaries relative to E56 ovaries and a downregulation of meiotic pathways in PND28 ovaries relative to PND8. To better contextualize the expression patterns, we also showed the union of all the genes that contributed to meiosis-related gene expression from our GSEA analyses (see R1.Q12). We found that 12 out of 20 qPCR genes clustered with the meiosis related genes that peaked in the NMR RNAseq data between PND1-PND8.

Page 9; 214-217: “We also completed a time-series differential expression analysis, which identified several genes of interest that were down-regulated genes across age (e.g., Pouf51, Top2a; Supplementary Figure 5, Supplementary Table 12), but no obvious ovarian development pathways were highlighted by GSEA (Supplementary Tables 14-15).”

b) To interpret the results with respect to known gene expression alterations during ovary development in other species we looked for comparable human and mouse data. While non comparable human data was found, we were able to use publicly available mouse ovarian gene expression data from Cardoso-Moreira et al., 2019 (DOI: [10.1038/s41586-019-1338-5](https://doi.org/10.1038/s41586-019-1338-5)). From this dataset we carefully selected 5 mouse ovarian development timepoints (E10.5, E13.5, P3, P14 and P28) that are most comparable to the naked mole-rat timepoints presented in this study. We then performed a comparative analysis by taking the union of the genes that contributed to the meiotic/gamete GSEA results from the E56 vs. PND1 comparison together with the selected qPCR genes (Figure 4B). This analysis further highlights the differences between NMR and mouse.

Page 9, Line 219-226: *“We next investigated how meiotic gene regulation during ovarian development compares between NMR and mice. To curate NMR ovarian developmental gene list, we combined the 76 genes identified from meiotic pathways in the GSEA analysis with the 20 genes profiled by qPCR, resulting in 65 genes after overlapping genes were removed (Figure 4a). We compared the expression of NMR meiotic genes to mice aged E10.5, E13.5, P3, P14 and P28³⁴. Indeed, most meiotic genes exhibited highest expression at E13.5 in mouse and postnatally in NMR (Figure 4b), highlighting the unusual postnatal upregulation of meiotic genes in during postnatal ovarian development in the NMR.”*

R1. Q13: Related to Supplementary Figure 5. The authors state: “... the first clear NOBOX signal by IF was observed at P15 (Supplementary Figure 5).” There is no panel for P15 in this Figure. Please complete the representative images and perform quantitation as in Figures 2 and 4, then state number of animals per group.

We apologize for this mistake, P15 NOBOX staining now is included.

Minor points

1. Related to Supplementary Figure 3
Please provide quantitation and statistics for VASA+/STRA8+ as done in Figures 2 and 4.

In the revised manuscript, we included the quantification of VASA+STRA8+ as well as the statistical analysis in supplementary Figure 7, lines 234-242.

2. Correct mislabeling in Figure 4 legend: “... and cleaved-caspase 1 (C-PARP1).”
Mislabeling is now corrected, thank you.

3. Correct typo in Figure 6 reference: “...the ovaries from a 10-year-old NMR (Figure 6c-e).”
Corrected.

4. Explain why P15 has less VASA+ cells than P8 AND P15? If it is because TUNEL is highest in P15, then KI67+ cells should be enhanced in P28 vs P15 and P90?

We observed an increased amount of TUNEL+ cells at P15 that is abruptly reduced at P28. However, the number of VASA+KI67+, VASA+pHH3+, and SOX2+KI67+ at P28 are similar to those observed at P15. As the reviewer indicated, we were expecting to see an enhancement of mitotic cells; however, the fact that the numbers are similar suggest that these cells will be able to maintain the germ cell population.

5. Related to Figure 5. While the prophase I chromosome spreads are impressive, their discussion in the text is insufficient for understanding. What is leptoneuma, zygonema etc. (stages of prophase I), why are they different, and how can this be quantified?

We include now in the text a section explaining the substages of the meiotic prophase I, indicating the differences between substages (lines 253-294). We evaluated different ages to confirm that at these developmental times, the age is not directly affecting the meiotic progression. Proportion of cells at the different substages of meiotic prophase I can be determined using meiotic spreads. Whereas meiotic spreads are a fantastic method to evaluate prophase dynamics, they are limited in their ability to evaluate the proportion of cells at each substage. Factors such as the number of cells per slide, how spread out are the chromosomes and the skill level of the user contribute to this methods limitation for assessing proportions. Other methods could be used for this purpose, such whole-mount staining, but owing to the massive amount of germ cells in the NMR ovaries and how tightly packed they are, it would be very difficult to perform a proper and reliable analysis of this sort.

6. gammaH2AX is not different between P8 and P28 as per Supplementary Figure 4a, but where are the error bars in Supplementary Figure 4b?

In the edited figure (now Supplementary Figure 8), error bars were added.

7. The conclusion presented for result from Figure 5c/d is unsatisfactory: Why has P28 still quite similar MLH1 foci number to P8, while the germ cells (VASA+, Figure 2d) are drastically different? And when REC8+VASA+ cells mark oocytes, and Figure 4d shows presence of oocytes throughout all time points with a similar pattern as total germ cell (Figure 2d), what is the relevance of the Figure 5d result?

All germ cells from late migratory primordial germ cells are positive for VASA, this includes oocytes during meiotic prophase. Our counts only include VASA+ cells before they progress to secondary follicles, and some of the variation in numbers is due to counting different germ cell types. In the revised version of the manuscript, we clearly state this (lines 245-248). One of the objectives of this project was to show that all the critical events of the meiotic prophase I occurred postnatally, which we've accomplished via Figure 5 d-g. All of the images in Figure 5 d-g are now from P8 ovaries, and we've removed the P28 images from 5 f and g, because they were potentially a source of confusion. Explaining the relevance of the difference in MLH1 foci between P8 and P28 is beyond the scope of this manuscript. Additionally, variation in the number of MLH1 at several more postnatal ages is one of the principal objectives of an ongoing study of meiosis in NMRs. To answer your question about MLH1 numbers being quite similar at P8 and P28; this indicates the processes of pairing-synapsis and recombination are happening in a similar fashion at both ages.

Reviewer #2 (Remarks to the Author):

Review of Brieño-Enriquez et al.

Summary

In the vast majority of mammals such as mouse, primates and humans, females are born with a reserve of follicles (oocytes and their surrounding cells) that deplete over their life span. Once this reserve is depleted, females reach reproductive senescence- in human this is known as menopause. It is therefore of interest to understand ovarian aging in different species. Ovarian aging is characterized by hormonal dysregulation, such as decreased sex steroids. A second layer of complexity is added by fertility rates, which are usually flat with female age- meaning that across a population, females reproduce until they die (or undergo menopause). Like the vast majority of mammals, naked mole rats (NMR) have a flat rate throughout their lifespan. What is different about NMR is that they live for decades and yet have high fertility rates until they reach their end of life span. In inbred mouse (in captivity), litter sizes ('fertility') decline after 6-9 months. It is therefore of interest to understand how litter sizes and fertility rates are maintained for three decades in the NMR.

Overall assessment: In this study, the authors provide a cytological characterization of germ cell development in the naked mole rat (NMR) ovary during the postnatal stages (P1 to P90- 3 months). They follow the dynamics of total germ cells (VASA+) from three animals at P1, P5, P8, P15, P28 and P90 and conclude that the ovarian pool is established in the post-natal ovary, as opposed to the pre-natal ovary as is seen in mice and humans. They further propose that the prolonged fertility in the NMR may be due to the persistence of PGCs/OSCs into later ages. This difference in timeline and fertility makes the NMR an interesting tool for studying early oogenesis and female reproductive aging.

We want to thank the reviewer for all the comments and constructive criticisms.

Major comments:

R2. Q1: PGC expansion in the postnatal ovary. One of the major strengths is the meiotic analysis conducted. The conclusion that meiotic prophase is taking place in the perinatal ovary is based on strong evidence from the Cohen lab. In contrast, the conclusion that naked mole rats expand their PGCs postnatally hinges on Figure 1, where the data and their statistical analyses are not clear.

In the revised manuscript the figures include a clear explanation of the statistical analyses performed, and we also include new results in Figures 1, 2, and 7 that support the expansion of the PGCs postnatally.

R2. Q2: In Figure 1a-c, the authors count total germ cells (VASA+) from three animals at P1, P5, P8, P15, P28 and P90. At P8, the three animals appear to have a higher number of total germ cells than at P5 and the authors conclude that there is an increase in PGCs/GCs around P8. However, the comparison of P8 with P1 does not yield a 'significant' p-value. Furthermore, there seems to be some technical issues with the ovary fixation at the critical P5- the H&E in Fig 1a doesn't look good. Is this just this specific section? Problems with fixation would also influence antibody staining of P5 in Fig. 1b, where the VASA staining at P5 looks as if the antibody did not

penetrate the tissue properly, causing an underestimate and could provide a rational explanation of the apparent low count in P5 compared to P8. More fundamentally, with only three animals per stage (P1, P5 and P8), this conclusion should be further supported- the data are not clear from these low numbers.

The count of total germ cells VASA+ in Figure 1 now includes counts of 6 ovaries at each age, enhancing the robustness of our results. We include a new representative image of P5 ovaries stained with H&E and VASA+.

R2. Q3: The assumption that VASA+ cells increase at P8 has downstream ramifications on the inference that pH3 and Ki67 markers are proliferating cells. These markers would also be present in meiotic cells, as VASA. It is therefore questionable whether further PCG expansion occurs in the postnatal ovary or whether it has all occurred in the fetal ovary.

In response to the reviewer's comment, we performed staining and analysis for VASA+Ki67+ and VASA+pHH3+ cells, as well as SOX2+Ki67 cells at all ages (P1 to P90). Quantification of these cells show that the numbers of VASA+Ki67+ and VASA+pHH3+ cells were high at P1 and are then reduced at P5. This suggests these cells might have started the mitotic process *in utero*. However, the increase at P8 would suggest new cells are actively dividing. We also observed the number of SOX2+Ki67+ cells at P5 is greater than at P1, which also supports the idea of the postnatal expansion. Lines 124-137.

R2. Q4: Finally, a basic tenet of stem cells is that they should be capable of proliferating *in vitro*. This should be demonstrated for the PGCs/OSCs (Sox2) that the authors suggest are oogonial stem cells.

In the revised manuscript we performed *in vitro* cultures of P5 ovaries for 5, 10, 20, 30, 60, 90 and 120 days (Figure 2 and Supplementary Figure 3). We treated the cultures with EdU at day 8 of culture and fixed at day 10. Fixed cells were stained for VASA, SOX2, DAPI and EdU. We observed VASA+ that had incorporated EdU, which shows some germ cells are actively dividing. Only a small subset of EdU+ cells were also SOX2+. Importantly, the fact that SOX2 positivity was limited to a small subset of these cells would be consistent with asymmetric division.

R2. Q5: Taken together, the data presented support the conclusion that meiotic prophase takes place postnatally, but do not convincingly demonstrate that postnatal PGC/OSC expansion occurs.

Thank you for your comments, they helped us to perform the proper follow up of the PGC expansion *in vivo* and *in vitro*. In fact, data supporting PGCs expansion is included in revised Figures 1 and 2, as well as Supplementary Figures 1 and 2, which support the expansion of PGCs *in vivo* and *in vitro*. Lines 172-197.

R2. Q6: Fertility preservation throughout life span. It is unclear how the authors are suggesting NMR preserve their fertility throughout life. In the model (Figure 6), it looks as if they suggest NMR establish their entire reserve during the first 3 month of their perinatal life. However, in the text they write that oogenesis could persist for many years (line 177). What is the authors' model? The ambiguity might in part be due to that their major data stop at P90 and they have a single ovary from a 10-year old female presented in Figure 6, coming after their model. The staining shows that germ cell nests (VASA positive) appear to be present in a 10-year old ovary. They further use this single ovary to suggest that 10-year old NMR has 5-times more germ cells than

mouse at 2 years (line 179). This last statement should either be removed or further data provided. A single data point cannot be taken as a given for an entire species (or colony). The manuscript would be much improved if the authors communicate clearly the various explanations for their observation- the VASA+ cells could be senescent, they could also be proliferating and initiating oogenesis throughout life. Again, demonstrating that these cells are capable of proliferating, a basic tenet of stem cells, should be demonstrated. At the very least far characterization and co-staining (e.g. SOX2) as well as an increased number of ovaries even from younger females (6 months, 1 year) would be needed to make firm conclusions about potential PGCs/OSCs. At the very least, one cannot conclude that NMRs maintain their fertility by continuous OSC activity throughout their life.

Originally, we developed the NMR model after studying ovaries up to P90 days; however, in the revised manuscript we have included data from P6MO, SUB-3yr and Ex-SUB-3yr females (Figure 7). For ovaries from these animals, we performed analyses of total germ cells (VASA+), as well as VASA+SOX2+, VASA+Ki67+ and VASA+pHH3+ cells. We also showed the presence of Ki67+SOX2+ cells in Ex-SUB-3yr ovaries and germ cell expansion *in vitro*. These results support the concept that the establishment of the ovarian reserve in NMRs is not limited to the first 3 months. We also include in Figure 8b and c images of germ cell nests in 3-year-old subordinate, 3-year-old reproductively active, 6-year-old subordinate, 6-year-old queen and 10-year-old subordinate females. The presence of germ cell nests in adult NMR ovaries could be explained by delayed breakdown of nests that formed earlier. However, based on our overall results, we think it's plausible that oogenesis persists up to 10 years of age. We hope to expand our analyses to 20- and 30-year-old subordinates and queens, but to date, we haven't had access to ovaries from NMRs of these advanced ages. Lines 315-355.

R2. Q7: Statements in the manuscript. The entire manuscript would benefit from removing statement that focus on how different naked mole rats are compared to other mammals and making them an example that postnatal OSC expansion exists. An example is the last sentence of the Abstract that germ cell proliferation occurs in the postnatal mammalian ovary. This extrapolation is not only inaccurate as many species do not contain OSCs.

We modified our comparative statements in the revised manuscript, but we've selected some to remain because of their importance to some readers. This is justified because some readers might need/want a reference to a model that is well established, such as mouse, but also because other reviewers requested us to compare our RNAseq data from NMR ovaries to RNAseq data from mouse ovaries. We would also like to mention that we intentionally did not use the term "OSCs", and favor PGCs since BLIMP1 positivity is unique to this cell type.

R2. Q8: There are several factual statements that are unclear in the Introduction and Discussion. This includes when females are fertile. Line 52-54 state no decline into their third decade of life, whereas line 60, says that they are fertile across their three decades-long lifespan. These two statements seem at odds with each other.

We unified the statements and now we mention that they are fertile during the entire lifespan. Lines 75-79

R2. Q9: There are several statements that need to be changed. For example, although women have lost 90% of the follicles by 30, they are still fertile- in fact, it's the peak for fertility in women. What is meant is that women have an extensive post-reproductive lifespan (post-menopause). These are but a few examples.

We edited some of these statements, now the sentence says “The queen’s protracted fertility is very relevant since reproductive longevity is based on the fact that female mammals harbor a finite number of oocytes that progressively decreases with age”,lines 80-82.

R2. Q10: Statistics. There were many examples where statistical comparisons had been carried out and the p-values but not effect sizes nor power calculations shown. Neither is it clear whether p-values had been adjusted when conducting multiple comparisons. It was difficult to evaluate objectively the biological significance in several of the experiments, especially in Figure 1.

We added a biostatistician to our team who has performed new analyses and re-evaluations of the data. This allowed us to develop some more compressible ways to depict the data. Now all figure legends include the method(s) used, the results of the model of multiple comparisons, and the p values of the comparisons.

Minor comments:

1. Figure 1. It’s difficult to appreciate that the ovaries grows with the scale bar at different sizes Figure 1a: the H&E staining of the P5 looks strange. Is this what you typically see at that stage? Please see my major comment.

The figure has been revised as suggested. The photomicrograph was replaced with a more representative image.

2. Figure 3 was poorly utilized. Information from the RNAseq analysis was not discussed in the paper. All of the expression data discussed was from supplemental figure 3; Moving figure S3 into the main text to replace the current figure 3 might make more sense.

For new Figures 3, 4 and supplementary figure 4, we re-analyzed the data following this reviewer’s advice, as well as the advice of other reviewers. In the revised manuscript we clearly describe the new methods used in each analysis, and the new data are properly described and discussed in the manuscript. We described in detail the data handling in the answers to reviewer #1.

3. It would help the readability of the manuscript to tone down the differences with other rodents throughout.

Some references to other rodents are included, but we have followed the reviewer’s advice and toned down our comparisons.

4. There are many repetitions that should be removed. E.g. line 60 has already been explained in the beginning of the paragraph.

Duplications and repetitions have been edited as suggested.

5. Please remove ‘Interestingly’, ‘remarkable’.

Text was heavily edited, and ‘interestingly’ and ‘remarkable’ were eliminated from the manuscript.

6. Line 196: the laboratory. Which one(s)?

In the methods section, we now stated that the ovaries were collected at the Holmes lab at the University of Toronto and then sent to the Brieño-Enriquez lab at MWRI, University of Pittsburgh (lines 410-411). And line 571, Postnatal day 5 ovaries were collected in PBS at 4°C in the Brieño-Enriquez laboratory.

7. Line 261: Total RNA isolation. Please specify what tissue, how much, and what cell type composition this tissue would be expected to have.

We performed RNA isolation from whole ovary for each one of the ages indicated. Based on our staining, counts and analyses, the predominant type of cells are germ cells from P1 to P28, and at P90 the germ cells are at different stages of growth and development and are accompanied by increased numbers of stromal and follicular cells. Lines 475-486

8. Line 333: I think Attila Toth's name is misspelt (if that is who you are thanking).

Thank you for catching this, we have corrected his name.

9. Some of the figures are mentioned out of order. For example, the TUNEL assay in figure 4d and e was found in the text after all of the meiosis analysis.

We apologize for these mistakes; these errors have been corrected in the revised manuscript.

10. There were several points where the wrong figure was listed. There were several points in the text where figure 3 was references, but the text was discussing figure 4.

In this new version of the paper, the figure mismatches have been corrected.

11. The paper would benefit from a deeper discussion on the use of this model organism in the field. The final paragraph was a nice summary of the study, but it's unclear whether much can be extrapolated.

Now we include in the discussion more on the use of the NMR model in the field of aging and reproduction and how it could be translated to human health. Lines 357-396.

Reviewer #3 (Remarks to the Author):

The manuscript presents a characterization of oogenesis in female naked mole rats. The main findings of the manuscript are that germ cells remain mitotically active at least for 8 days after birth, and that meiosis is initiated in germ cells only 8-28 days after birth. These properties of naked mole rat oogenesis contrast oogenesis in mice and humans, where germ cells stop mitotic divisions and complete meiotic prophase in foetal ovaries. The manuscript also reports germ cell nests in 10 years old animals, which raises the possibility that germ stem cells or mitotic germ cells persist well into adulthood in naked mole rat. However, the cell cycle status of these adult germ cells is not determined, thus it is not clear if they are mitotic or meiotic cells, and/or they are in a resting phase having past through meiotic prophase. The manuscript also finds an exceptionally large ovarian reserve in 8 days old naked mole rats, which may provide explanation for protracted fertility in this species.

The presented data provides sufficient evidence for the main conclusion that there is postnatal germ cell proliferation and meiosis initiation in female naked mole rats. This finding is interesting to the reproduction biology field and is consistent with other peculiar features of naked mole rats, which probably, co-evolved with their eusociality. However, I am not certain that the depth and the importance of the discoveries of the manuscript justify access to the wide readership of Nature Communications. The manuscript may fit better a specialized journal, such as Biology of Reproduction or Developmental Biology. This is because the main findings do not reveal a very surprising or a generalizable phenomenon. The main discovery is that the developmental timing of meiosis initiation is shifted to a slightly later age in naked mole rats as compared to humans or mouse. Nonetheless, the general pattern of naked mole rat oogenesis seems similar to oogenesis in other mammals: intense mitotic proliferation of germ cells takes place in immature animals, proliferation seem to cease or nearly cease in juveniles and germ cells go through meiotic prophase in immature animals, hence a finite oocyte pool is generated that seem to support fertility throughout mature life. The slightly delayed timing of this developmental pattern in a highly specialized animal species is not very surprising given the already reported differences between oogenesis in humans and mice, which are summarized in figure 6. Thus, while the manuscript represents a descriptive study of oogenesis in a species that is interesting for longevity and regeneration studies, it does not uncover a truly novel or surprising concept. Beyond the above general concern, I would have several suggestions for improvements in the manuscript.

We want to thank the reviewer for the comments and constructive criticisms.

R3. Q1: The presentation of the ovarian transcriptome analysis in figure 3 is not very informative as it only shows that the transcriptome changes during development, which is not a very revealing conclusion. Including this data to the main figure would be justified only if the manuscript presented more analysis of RNAseq data. For example, the figure could present changes in the expression of various marker genes for PGCs, mitotic germ cells and meiotic germ cells matching the qPCR analysis in supplementary figure 3.

In the revised manuscript we re-analyzed the results from RNA-seq and prepared 4 new figures that evaluate the differentially expressed genes in the developing ovary (Figure 3 ; compare ovarian development of NMR to mouse (Figure 4), and we contrasted the RNA-seq to qPCRs (Figures 3 and 4).

R3. Q2: The manuscript present data on mitotic proliferation 8 days after birth and meiotic prophase progression only from 8- or 28-days old animals. It is somewhat ambiguous in the text, but my understanding is that all immature stages were tested for mitotic proliferation, but

proliferation was seen only in P8 animals. In contrast, meiotic prophase was only looked at in P8 and P28 animals. These analyses are incomplete without testing other developmental stages including 90days old or mature animals. Are germ cell proliferation and meiotic initiation take place throughout the life of naked mole rats or they are restricted to immature stages?

In the revised manuscript we include the analysis of the mitotic proliferation markers (Ki67+VASA+ (Figure 1) and pHH3 (supplementary Figure 1) from P1- P90. These analyses show that dividing germ cells are present at all these ages. We also evaluated the dynamics of SOX2+Ki67+ cells from P1 to P90 (Figure 2). In Figure 5, we include the evaluation of VASA+REC8+ cells from P1 to P90. In the revised manuscript, we also include ovaries from 6MO, SUB-3yr and Ex-SUB-3yr female, and in these animals, we quantified the total number of germ cells (VASA+), as well as the VASA+SOX2+ cells, and markers of proliferation. We observed the presence of VASA+SOX2+ cells in all three cohorts. In fact, we were able to identify VASA+Ki67+ cells and VASA+ SOX2+ cells in all of them; however, they were significantly increased in the reproductively active females (Ex-SUB-3yr). We observed STRA8 and REC8 signal in older animals, but we did not perform meiotic spreads. We agree that a full meiotic prophase profile at all the ages will be beneficial, however for this project the main purpose was to show that in fact meiotic prophase is happening and all the critical steps are properly placed. In fact, we are working on the study of how age regulates meiotic prophase I in NMRs, but the work is ongoing and out of the main scope of this manuscript.

R3. Q3: Further, I may have missed it, but does the manuscript show histology of ovaries from a 10-year-old breeder queen or a non-breeding worker in Figure 6. To get an accurate picture of how oogenesis is adapted to fit the unique eusociality of naked mole rats, it would be essential to examine the oocyte pool in both adult workers and queens. Is there a difference in the oocyte pool size in adult queens and workers? Is there differential mitotic germ cell proliferation and germ cell pool replenishment in adult workers and queens? Apparent cessation of mitotic divisions in immature animals that are destined to become sexually inactive workers might not reflect the status of germ cells in adult queens. Thus, are there dividing mitotic germ cells in adult queens? Finally, are there significant numbers of germ cells that initiate meiosis in adult queens as compared to workers? These are all very important questions that are left unanswered by the manuscript, which severely limits the depth of the conclusions and the value of the manuscript.

The reviewer raised very important points. In the revised manuscript, we also include ovaries from 6MO, SUB-3yr and Ex-SUB-3yr female. We evaluated the total number of germ cells (VASA+), as well as the VASA+SOX2+ cells, and markers of proliferation (Figure 7). We observed the presence of VASA+SOX2+ cells in all of them, however the number of VASA+SOX2+ cells was statistically significant higher in the Ex-SUB-3yr. As we mention on R3Q2 we were able to identify VASA+Ki67+, VASA+pHH3+ and VASA+SOX2+ cells in all of them; however, they were significantly increased in the reproductively active females (Ex-SUB-3yr). We detected cells positive for STRA8 and REC8 in adult animals, however how queens regulate the meiotic entry compared to subordinates of the same age it is part of an ongoing project.

R3. Q4: Connected to point two, analysis of germ cell nests from the 10 years old adult is incomplete. It is not clear what cell cycle status germ cells have in the nests. It is also not clear if germ cell nests are present in both queens and workers or only in workers assuming that the presented image is from a worker. It would not be surprising if folliculogenesis was delayed in non-breeder workers as compared to breeding queens.

We agree with the reviewer; the picture of a 10-year-old female's ovary was from a non-reproductive, subordinate NMR, and that can lead to confusion. We now include data from 3-year-old subordinated female (SUB-3yr), 3-years-old reproductively activated female (Ex-SUB-3yr), 6-year-old subordinate female, 6-year-old queen and 10-year-old subordinate female in Figure 8. In all the cases we observed presence of germ cell nests, independently of the age or the reproductive status. However, as the reviewer pointed out, in reproductively activated females we observed nests, primordial, primary, secondary, antral follicles, meanwhile in subordinates we did not observed later stages of folliculogenesis.

R3. Q5: It would improve the manuscript if there was more discussion about the possible interplay between the eusociality and the developmental pattern of oogenesis in naked mole rats. In particular, how the life cycle of the naked mole rats permitted and/or necessitated changes in the timing and regulation of oogenesis in worker and breeder naked mole rats as compared to their rodent relatives.

We agree with the reviewer, in the revised manuscript we include a paragraph where we talk about this very interesting interplay between the eusociality and the oogenesis lines 385-394). “Indeed, we propose that, the large ovarian reserve and potential for postnatal oogenesis that we report is presumably closely intertwined with the adaptations for remarkable longevity and eusociality in this species. Naked mole-rats have the largest reproductive skew among mammals (Faulkes & Bennett, 2021); while the vast majority of adult females will never breed, all need to be ready should the opportunity arise. The impact on direct reproductive fitness is huge for those females that attain and maintain queen status. Queens can produce 4-5 litters per year, averaging 13 pups per litter (Smith & Buffenstein, 2021) meaning a queen could conceivably produce thousands of offspring over her extreme reproductive lifespan. Thus, the evolutionary pressures on ovarian development and function in this species are particularly striking”.

REVIEWER COMMENTS

Reviewer #1 (Remarks to the Author):

The authors addressed all of my concerns

Reviewer #2 (Remarks to the Author):

Review:

The manuscript is improved and the data show improved quality and quantity for the conclusions. The conclusions have been strengthened significantly by addition of further females, which clearly has been challenging during covid. The data on the in vitro stem cells really significant value and I'd like to commend and thank the authors for doing this, especially during such difficult times. The authors have also improved the communication and have added clarifying information that will help the broad readership. I particularly appreciated the integration with life history of NMRs in understanding the scientific hypothesis.

I have a few comments that I believe would further strengthen the manuscript.

Line 58: studies in humans are often prohibitively expensive and ethically limited. I am really unsure what this statement is based on. Many original studies on human ovaries was done on material obtained from hysterectomies, using standard H&E staining. These studies aren't cited here and it's therefore very strange to see this statement. It's also unclear what is meant by ethically limited – this could be construed in many ways, but the original human ovarian studies have been as valuable as the one conducted here, which is by necessity also descriptive in nature. If the authors choose to leave in such comments, then they should be explicit and well-founded.

References: there are places where some studies are cited but others are not. E.g. line 124, where Ki-67 is cited as a proliferation marker but pHH3 isn't.

The referencing would also improve by placing relevant references with the organism or statement. E.g. line 120, mouse and human references are clustered together at the end of the sentence, implying they all refer to human. Specifically, the statement on the ovarian reserve at birth for human isn't quite correct. There is substantial debate about the number of follicles at birth in human – but one to two million are known; this is because the modelling of numbers show a wide range (from 250 000 til 2.5 million; e.g. Wallace and Kelsey, Human ovarian reserve from conception to menopause, PLoS one e:8772, 2010). Therefore, stating that 1.5 million germ cells is 5 times greater than humans at birth is an exaggeration. There is no need to potentially unfounded claims– the importance of this study and the value of NMR is clear.

Would you please consider citing original research? Although reviews are valuable, it would be good to recognize the papers that actually carried out the work. There are several examples of this throughout.

The statement on mouse and human oocytes and in utero (line 374). I don't think it's clear why NMR would be helpful as an aging model for human aneuploidy, given that they

replenish they replenish their reserve even in adult females. Furthermore, there are in vitro models of mouse oogenesis that will allow in vitro studies of meiosis and chromosome segregation (Hayashi and Saitou labs; Nature and Science papers).

My final comment is that although the exaggerated and at times, in my view, dramatic style has been toned down, there are still places where it detracts from the scientific quality. The initial sentence of the paper suggesting the OSCs are highly controversial has pretty much settled down and been debunked. The company selling the OSC treatment doesn't do so anymore, the polyclonal antibody used to purify OSCs was found to pick up MSCs, etc. etc. I don't think for humans that OSCs are 'highly' controversial'.

Reviewer #3 (Remarks to the Author):

The manuscript has been greatly improved as compared to the previous version as a lot of new data has been included. Accordingly, the results show more convincingly that the meiotic prophase takes place in postnatal life. Nonetheless, the main issue remains. The presented data is still incomplete, therefore the main message of the manuscript is not sufficiently supported.

The main claim of the manuscript is that postnatal oogenesis is responsible for an exceptionally large ovarian reserve and protracted fertility in naked mole rats (NMR). Whereas the manuscript clearly demonstrates that both mitotic and meiotic phases of oogenesis occur after birth in NMR, the data do not show that the shift in the timing of oogenesis leads to enlarged oocyte pool, protracted fertility, or new oocyte production in adults. Thus, the statement in the title is misleading. Similar misleading statements are made in the abstract and the body of the text (see lines 38-40, 357-358, 380-382, 389-391). Along the same line, the summary of oogenesis in figure 8a hints meiosis initiation at more advanced developmental stages than what is supported by the data.

As it stands the manuscript presents very convincing evidence that there is a mitotic expansion of oogenic cells soon after birth (1-8 days), followed by a major asynchronous wave of meiosis that is strongly attenuated by 28 days after birth. Whereas both mitotic primordial germ cell (PGC)-like cells and meiotic cells are reported up to 90 days post-birth, their number is greatly reduced as compared to perinatal (1-8days) NMRs. Therefore, one can conclude that most oogenic mitotic divisions take place perinatally and that most oogenic cells complete meiotic prophase in adolescent/juvenile NMRs. Thus, the currently presented data is consistent with the scenario that a large oocyte reserve is created soon after birth in immature animals, and that this oocyte reserve is used throughout the lifetime of NMRs without replenishment in adults.

Whereas the developmental timing of oogenesis seems to be delayed slightly in NMR as compared to mice and humans it is not clear if and how the shift of bulk oogenesis to perinatal/adolescent timepoints is relevant for generating a large oocyte pool or protracted fertility in NMR. In contrast to oogenesis in juveniles, if oogenesis existed in adults, adult oogenesis would be highly relevant for protracted fertility in NMRs. However adult oogenesis is incompletely analyzed in the manuscript; key experiments are missing.

Relevant to the above point, the manuscript presents very interesting data on PGC-like cells in adults, including reproductively activated NMRs (Ex-SUB 3yr). The authors find that the number of mitotically active PGC-like cells is significantly elevated in Ex-SUB 3yr naked mole rats as compared to subordinate females, which are not reproductively active. These observations are very significant as they raise the possibility that there is a renewed production of functional oocytes in reproductively activated adults. However, it is not examined if reproductive activation significantly elevates total oocyte numbers over longer developmental times. Crucially, it is not addressed either if the dividing PGC-like cells give rise to functional oocytes that undergo meiotic prophase in adult Ex-SUB 3yr naked mole rats.

In their response to reviewers, the authors state that an examination of meiosis is underway in adult reproductively active NMRs, and that the analysis of meiotic initiation in adults is beyond the main scope of the current manuscript. Whereas I understand the time constraints and the difficulty of working with NMRs, I don't have the same opinion as the authors on this point. The self-proclaimed aim of the manuscript is to address how protracted fertility is maintained in NMRs. This aim is also reflected by the title. Thus it is crucial to address if new divisions of PGC-like cells in adults result in oogenic cells that undergo meiosis. It is not sufficient to show that there are mitotically active PGC-like cells as their functionality is uncertain. The presence of PGC markers does not mean that the corresponding cells are able to undergo meiosis and produce fertilizable oocytes. Therefore, it is essential for the message of the paper to test if oocytes that initiate meiotic prophase are present in adult Ex-SUB 3yr or 6-year-old NMRs.

To this end, meiotic spreads have to be examined and/or a marker of meiotic recombination/initiation (e.g. *HORMAD1*) has to be detected on sections. Detection of *REC8* in sections is not sufficient because *REC8* also marks oocytes after meiotic recombination is completed, hence *REC8* staining of sections informs us about the total number of cells that entered and/or completed meiotic recombination as opposed to the number of cells that are undergoing meiotic recombination.

The manuscript is somewhat ambiguous about data concerning meiosis initiation in adults. In lines 376 -379 it is hinted that *MLH1* focus numbers/*CO* numbers differ in oocytes that enter meiosis at distinct developmental stages. It is not clear what developmental stages have been looked at. Does this statement mean that the authors have already analyzed prophase stage oocytes in reproductively active adult females? I fully agree with the authors that an analysis of *MLH1* focus counts in different ages is beyond the scope of the current manuscript, but if there is data showing the existence of early prophase stage oocytes in adults the data should be included in the manuscript.

Additional points:

Figure 8a indicates that meiosis initiation peaks around 28 days after birth and it persists beyond one year of age. The transcription data and *REC8* counts do not support this conclusion. According to the measurements, meiosis initiation may peak around 8 days after birth and not at 28 days. I note that interpretation of the data is difficult as the transcription and the number of prophase stage cells are not normalized to the total number of germ cells. Hence, it is difficult to tell when the highest fraction of germ cells initiates meiosis. The percentage of germ cells that entered meiotic prophase should be measured. Further, there is no data showing meiotic prophase occurring in animals that are older than 90 days, hence it should be somehow indicated that meiosis initiation may or may not occur in adults.

Figure 5: It would be useful to show the percentage of HORMAD1 or gamma H2AFX positive VASA+ cells. REC8 marks prophase cells but it is not suitable to judge how many cells enter meiotic prophase because REC8 is present in early prophase, resting oocytes and oocytes that reenter meiotic divisions before fertilization. In contrast HORMAD1 and gamma H2AFX are transiently present in oocytes that are active in recombination at the beginning of prophase. Hence counting VASA + cells that are positive for HORMAD1 and gamma H2AFX would allow the authors to identify the developmental stage where the highest fraction of cells enter meiosis. This would be useful for the main message of the manuscript, see also comment to figure 8a.

A minor point: Please specify what the different colours mean in the graph showing MLH1 focus quantification.

Line 377: There is a typo, "a reduction" is repeated.

Our preliminary data suggests that with age
377 there is a reduction a reduction of MLH1 foci in oocytes that entered into meiosis later
during
378 development

REVIEWER COMMENTS

Reviewer #1 (Remarks to the Author):

The authors addressed all of my concerns

Thank you.

Reviewer #2 (Remarks to the Author):

Review:

The manuscript is improved and the data show improved quality and quantity for the conclusions. The conclusions have been strengthened significantly by addition of further females, which clearly has been challenging during covid. The data on the in vitro stem cells really significant value and I'd like to commend and thank the authors for doing this, especially during such difficult times. The authors have also improved the communication and have added clarifying information that will help the broad readership. I particularly appreciated the integration with life history of NMRs in understanding the scientific hypothesis.

Thank you, we really appreciate your kind comments.

I have a few comments that I believe would further strengthen the manuscript.

R2.Q1. Line 58: studies in humans are often prohibitively expensive and ethically limited. I am really unsure what this statement is based on. Many original studies on human ovaries was done on material obtained from hysterectomies, using standard H&E staining. These studies aren't cited here and it's therefore very strange to see this statement. It's also unclear what is meant by ethically limited – this could be construed in many ways, but the original human ovarian studies have been as valuable as the one conducted here, which is by necessity also descriptive in nature. If the authors choose to leave in such comments, then they should be explicit and well-founded.

Following the reviewer's comment, we removed the sentence.

R2.Q2. References: there are places where some studies are cited but others are not. E.g. line 124, where Ki-67 is cited as a proliferation marker but pHH3 isn't.

Thank you, we've now included references for pHH3 as follows: Kim, B. et al. Primordial germ cell proliferation is impaired in Fused Toes mutant embryos. Dev Biol 349, 417-426, doi:10.1016/j.ydbio.2010.10.010 (2011); Anderson, R. A. et al. Activation of the aryl hydrocarbon receptor by a component of cigarette smoke reduces germ cell proliferation in the human fetal ovary. Mol Hum Reprod 20, 42-48, doi:10.1093/molehr/gat059 (2014). Lines 121-123

R2.Q3. The referencing would also improve by placing relevant references with the organism or statement. E.g. line 120, mouse and human references are clustered together at the end of the sentence, implying they all refer to human.

Following the reviewer's advice, we separated mouse and human references to avoid confusion.

R2.Q4. Specifically, the statement on the ovarian reserve at birth for human isn't quite correct. There is substantial debate about the number of follicles at birth in human – but one to two million are known; this is because the modelling of numbers show a wide range (from 250 000 til 2.5 million; e.g. Wallace and Kelsey, Human ovarian reserve from conception to menopause, PLoS one e:8772, 2010). Therefore, stating that 1.5 million germ cells is 5 times greater than humans at birth is an exaggeration. There is no need to potentially unfounded claims– the importance of this study and the value of NMR is clear.

We agree with the reviewer about the number of follicles at birth in human, with substantial variability among publications. As the reviewer mentioned, the relevance of our results is clear, so we removed this sentence from the revised manuscript.

R2.Q5. Would you please consider citing original research? Although reviews are valuable, it would be good to recognize the papers that actually carried out the work. There are several examples of this throughout.

We changed some of the references to achieve a balance between reviews and original papers, while also maintaining the number of references close to the journal's guidelines.

R2.Q6. The statement on mouse and human oocytes and in utero (line 374). I don't think it's clear why NMR would be helpful as an aging model for human aneuploidy, given that they replenish they replenish their reserve even in adult females. Furthermore, there are in vitro models of mouse oogenesis that will allow in vitro studies of meiosis and chromosome segregation (Hayashi and Saitou labs; Nature and Science papers).

***In vitro* gametogenesis is a powerful tool to study development, and as mentioned by the reviewer. We agree that the replenishment of germ cells might seem to limit the NMR as an aging model to study aneuploidy. However, our preliminary data suggest that with age there is a reduction in MLH1 foci in NMR oocytes that entered meiosis later during development. Of course, more studies are needed support our hypothesis and to establish mechanisms and draw conclusions.**

R2.Q7. My final comment is that although the exaggerated and at times, in my view, dramatic style has been toned down, there are still places where it detracts from the scientific quality. The initial sentence of the paper suggesting the OSCs are highly controversial has pretty much settled down and been debunked. The company selling the OSC treatment doesn't do so anymore, the polyclonal antibody used to purify OSCs was found to pick up MSCs, etc. etc. I don't think for humans that OSCs are 'highly' controversial'.

Following the reviewer's comment, we edited the first sentence to "The adult mammalian ovary is devoid of germline stem cells. As such, female reproductive aging largely results from the depletion of a finite ovarian follicle pool." Lines 27-28.

Reviewer #3 (Remarks to the Author):

R3.Q1. The manuscript has been greatly improved as compared to the previous version as a lot of new data has been included. Accordingly, the results show more convincingly that the meiotic prophase takes place in postnatal life. Nonetheless, the main issue remains. The presented data is still incomplete, therefore the main message of the manuscript is not sufficiently supported. The main claim of the manuscript is that postnatal oogenesis is responsible for an exceptionally large ovarian reserve and protracted fertility in naked mole rats (NMR). Whereas the manuscript clearly demonstrates that both mitotic and meiotic phases of oogenesis occur after birth in NMR, the data do not show that the shift in the timing of oogenesis leads to enlarged oocyte pool, protracted fertility, or new oocyte production in adults. Thus, the statement in the title is misleading. Similar misleading statements are made in the abstract and the body of the text (see lines 38-40, 357-358, 380-382, 389-391). Along the same line, the summary of oogenesis in figure 8a hints meiosis initiation at more advanced developmental stages than what is supported by the data.

We want to thank the reviewer for the comments to improve our work. In the current revised version of the manuscript, we included the analysis of markers of meiotic prophase initiation in 6-month-old, 3-year-old subordinate and 3-year-old reproductively activated females (see R3.Q3 and Figure 8a-f). These results also allowed us to modify the model included in figure 8a, which is now Fig. 8g. The figure is now clearly labeled and shows the peak of meiotic prophase occurs at postnatal day 8 but extends to at least up to 3 years of age.

R3.Q2. As it stands the manuscript presents very convincing evidence that there is a mitotic expansion of oogenic cells soon after birth (1-8 days), followed by a major asynchronous wave of meiosis that is strongly attenuated by 28 days after birth. Whereas both mitotic primordial germ cell (PGC)-like cells and meiotic cells are reported up to 90 days post-birth, their number is greatly reduced as compared to perinatal (1-8days) NMRs. Therefore, one can conclude that most oogenic mitotic divisions take place perinatally and that most oogenic cells complete meiotic prophase in adolescent/juvenile NMRs. Thus, the currently presented data is consistent with the scenario that a large oocyte reserve is created soon after birth in immature animals, and that this oocyte reserve is used throughout the lifetime of NMRs without replenishment in adults. Whereas the developmental timing of oogenesis seems to be delayed slightly in NMR as compared to mice and humans it is not clear if and how the shift of bulk oogenesis to perinatal/adolescent timepoints is relevant for generating a large oocyte pool or protracted fertility in NMR. In contrast to oogenesis in juveniles, if oogenesis existed in adults, adult oogenesis would be highly relevant for protracted fertility in NMRs. However adult oogenesis is incompletely analyzed in the manuscript; key experiments are missing.

As we mentioned on R3.Q1 and later on R3.Q3, the new version of the manuscript includes the analysis of 3-year-old adults (subordinate and reproductively activated females). In these experiments, we were able to show entry into meiotic prophase in adults. The reviewer suggested we perform additional experiments involving NMRs older than 3 years of age to determine whether our observations are present in significantly older animals and in females that were kept for longer than 4 weeks post reproductive activation. Whereas these sorts of experiments are among our long-term research goals on NMR ovarian biology, they are well beyond the scope of this manuscript and would require years to complete. The NMRs in Briño-Enríquez lab are years away from reaching the desired target ages. And upon reaching the advanced target ages, we would want to reproductively activate these older females

and wait up to 12 months before interrogating their ovaries to satisfy our interests, as well as those of the reviewer.

R3.Q3. Relevant to the above point, the manuscript presents very interesting data on PGC-like cells in adults, including reproductively activated NMRs (Ex-SUB 3yr). The authors find that the number of mitotically active PGC-like cells is significantly elevated in Ex-SUB 3yr naked mole rats as compared to subordinate females, which are not reproductively active. These observations are very significant as they raise the possibility that there is a renewed production of functional oocytes in reproductively activated adults. However, it is not examined if reproductive activation significantly elevates total oocyte numbers over longer developmental times. Crucially, it is not addressed either if the dividing PGC-like cells give rise to functional oocytes that undergo meiotic prophase in adult Ex-SUB 3yr naked mole rats.

Following the reviewer's advice, we addressed the following question: Are there cells entering meiosis in the adult naked mole-rat ovaries? We analyzed NMR ovaries in 6-month-old, 3-year-old subordinate and 3-year-old reproductively activated females and stained for and quantified the number of cells positive for VASA+STRA8+, VASA+yH2AX and, VASA+HORMAD1+. Our results show the presence of VASA+STRA8+ cells in the three groups without any statistical differences among them (Figure 8a and 8b). However, when we quantified the numbers of cells positive for VASA+yH2AX and VASA+HORMAD1+ cells, we observed a statistically significant difference among 3-year-old reproductively activated females and both 6-month-old and 3-year-old subordinate females (Figure 8c-f, lines 345-359). No difference was observed between 6-month-old and 3-year-old subordinate females. These results indicate hundreds to thousands of cells can enter into meiosis at 6 months and 3 years of age in subordinate females. And perhaps most importantly, reproductive activation of the naked mole-rats at 3 years of age induces a statistically significant increase in the number of germ cells that are mitotically active, as well as the number of germ cells that are positive for markers of meiotic prophase initiation and progression (STRA8+, yH2AX and, HORMAD1+).

R3.Q4. In their response to reviewers, the authors state that an examination of meiosis is underway in adult reproductively active NMRs, and that the analysis of meiotic initiation in adults is beyond the main scope of the current manuscript. Whereas I understand the time constraints and the difficulty of working with NMRs, I don't have the same opinion as the authors on this point. The self-proclaimed aim of the manuscript is to address how protracted fertility is maintained in NMRs. This aim is also reflected by the title. Thus it is crucial to address if new divisions of PGC-like cells in adults result in oogenic cells that undergo meiosis. It is not sufficient to show that there are mitotically active PGC-like cells as their functionality is uncertain. The presence of PGC markers does not mean that the corresponding cells are able to undergo meiosis and produce fertilizable oocytes. Therefore, it is essential for the message of the paper to test if oocytes that initiate meiotic prophase are present in adult Ex-SUB 3yr or 6-year-old NMRs. To this end, meiotic spreads have to be examined and/or a marker of meiotic recombination/initiation (e.g. HORMAD1) has to be detected on sections. Detection of REC8 in sections is not sufficient because REC8 also marks oocytes after meiotic recombination is completed, hence REC8 staining of sections informs us about the total number of cells that entered and/or completed meiotic recombination as opposed to the number of cells that are undergoing meiotic recombination.

Following the reviewer comments, we performed the detection of the meiotic initiation/ recombination on histology sections (please see R3.Q3).

R3.Q5. The manuscript is somewhat ambiguous about data concerning meiosis initiation in adults. In lines 376 -379 it is hinted that MLH1 focus numbers/CO numbers differ in oocytes that enter meiosis at distinct developmental stages. It is not clear what developmental stages have been looked at. Does this statement mean that the authors have already analyzed prophase stage oocytes in reproductively active adult females? I fully agree with the authors that an analysis of MLH1 focus counts in different ages is beyond the scope of the current manuscript, but if there is data showing the existence of early prophase stage oocytes in adults the data should be included in the manuscript.

Analysis of meiotic spreads was performed initially on P8 and P28 animals, not in adults. As we mentioned on R3.Q3. the manuscript now includes the analysis of VASA+STRA8+, VASA+ yH2AX+ and VASA+HORMAD1+, histology sections from adults. Figure 8a-f and lines 345-359.

Additional points:

Figure 8a indicates that meiosis initiation peaks around 28 days after birth and it persists beyond one year of age. The transcription data and REC8 counts do not support this conclusion. According to the measurements, meiosis initiation may peak around 8 days after birth and not at 28days. I note that interpretation of the data is difficult as the transcription and the number of prophase stage cells are not normalized to the total number of germ cells. Hence, it is difficult to tell when the highest fraction of germ cells initiates meiosis. The percentage of germ cells that entered meiotic prophase should be measured. Further, there is no data showing meiotic prophase occurring in animals that are older than 90 days, hence it should be somehow indicated that meiosis initiation may or may not occur in adults.

We agree with the reviewer and the figure was edited to more accurately represent our data. We now include data from 6-month-old, 3-year-old subordinate and 3-year-old reproductively active females.

Figure 5: It would be useful to show the percentage of HORMAD1 or gamma H2AFX positive VAS+ cells. REC8 marks prophase cells but it is not suitable to judge how many cells enter meiotic prophase because REC8 is present in early prophase, resting oocytes and oocytes that reenter meiotic divisions before fertilization. In contrast HORMAD1 and gamma H2AFX are transiently present in oocytes that are active in recombination at the beginning of prophase. Hence counting VASA + cells that are positive for HORMAD1 and gamma H2AFX would allow the authors to identify the developmental stage where the highest fraction of cells enter meiosis. This would be useful for the main message of the manuscript, see also comment to figure 8a.

The previous version included supplemental material that included the quantification of VASA+STRA8+ cells in ovaries from P1 to P90 females (lines 228-232, and Supplementary Figure 7a). In this new version of the paper, we also include the total number of VASA+STRA8+, VASA+yH2AX+ and VASA+HORMAD1+ cells in animals at P6MO, SUB 3yr and Ex-SUB 3yr (Figure 8a-f and lines 345-359)

A minor point: Please specify what the different colours mean in the graph showing MLH1 focus quantification.

Individual colours indicate cells are from the same animal, which is now it is indicated in the figure legend. Lines 977-978

Line 377: There is a typo, "a reduction" is repeated.
Thank you, we corrected the typo.

REVIEWERS' COMMENTS

Reviewer #3 (Remarks to the Author):

The main conclusions of the manuscript are that (i) both mitotic germ cell proliferation and meiosis entry postnatally take place in naked mole rats (NMRs), and that (ii) de novo meiosis entry/neo-oogenesis occurs in adults, which is a unique reproductive strategy that supports maintenance of oocyte pools and fertility during the long reproductive life of NMRs. The previous version of the manuscript convincingly showed that a large cohort of germ cells enters meiosis soon after birth thereby generating a large oocyte reserve in juvenile NMRs. However, evidence was missing for de novo meiosis initiation in adults. The new version of manuscript includes new data that aims to fill this gap. To this end marker proteins of meiotic prophase, STRA8, gamma H2AX and HORMAD1, were detected in sections of ovaries of adult NMRs (note that "adult" refers to SUB 3yr and EX-SUB 3yr in this review). Whereas these experiments formally attempted to satisfy requests from the previous review round, they fell short of addressing the essence of the requests. Hence, the main problematic issue remains the same, de novo initiation of normal meiosis is not conclusively demonstrated.

In the previous review round, it was requested that a marker of meiosis initiation is detected in germ cells of adults to test if meiosis initiation takes place. HORMAD1 was suggested as a reliable marker and, in general, the use chromosome spreads was encouraged, because unique meiosis specific chromatin structures, axial elements and the synaptonemal complexes, which are marked by HORMAD1 or REC8 are much easier detected on nuclear surface spreads as opposed to sections. It is very dangerous to rely only on expression of marker proteins as opposed to their subcellular localization as a criteria for judging if meiosis is induced in germ cells. In the past, a reliance on protein and RNA expression data led to the meanwhile discredited notion of oogenic stem cells/meiosis in adult in humans.

Therefore, a detection of chromosome structures that are unique to meiosis are essential to test if a functional meiosis is initiated in adults.

The presented data does not reveal if meiosis specific chromosome structures normally form in germ cells of adult NMRs. The granular/pan-nuclear HORMAD1 staining patterns are surprising, as filamentous structures resembling REC8 patterns (fig 5) would be expected for an axial element component, such as HORMAD1, in meiotic prophase. The HORMAD1 staining patterns may have two possible explanations. (i) HORMAD1 is not localized on axial elements in oocytes of adult NMRs. Hence, the HORMAD1 staining shown in figure 8 may represent potentially abnormal expression of HORMAD1 in non-meiotic cells. (ii)

Nucleoplasmic HORMAD1 may mask filamentous HORMAD1 patterns in functionally normal meiotic cells in sections. To distinguish between these possibilities HORMAD1 should have been detected in chromosome spreads in oocytes of 3 years old NMRs as suggested in the previous review round. Further, as a control, HORMAD1 could have been detected in perinatal (15-28 days) sections to test if HORMAD1 staining produces filamentous patterns in sections at a stage where large fractions of oocytes enter meiosis. I note that It could not be predicted in advance if HORMAD1 staining on sections would or would not reproduce the filamentous patterns that are typical of axial element components such as REC8 (see figure 5, both sections and nuclear surface spreads). Therefore, the use of both chromosome spreads (for reliable detection of axial elements) and sections (for simplicity and reduced effort) were suggested in the previous review round.

A lack of analysis of subcellular localization of HORMAD1 is not the only problem. It is very worrying that HORMAD1 is present in much lower numbers (around 300 cell) of VASA positive germ cells than STRA8 or gamma H2AX (both are in the 7000 range, 20-25 fold more), which are also used as meiotic markers (fig. 8). Based on the developmental expression patterns of these markers in mice, HORMAD1 should be expressed in similar or

possibly higher proportions of meiotic cells than STRA8 or gammaH2AX. The discrepancies in the frequency of meiotic markers suggest that a functional meiotic programme is inefficiently, if at all, initiated in adult NMRs. Strong gamma H2AX accumulation can be observed also in meiotic cells that undergo apoptosis in mice, hence the data may indicate that germ cells attempt to initiate meiosis but fail and undergo cell death in adults. These considerations also highlight the dangers of relying on expression of marker proteins as opposed to their subcellular localization as a criteria for meiosis.

In short, in my opinion, the presented data argue against the hypothesis that functionally normal meiosis takes place in significant numbers in adult NMRs. At best, the data are inconclusive about meiosis in adult female NMRs. Therefore, further data should be included to clarify if meiosis specific chromosome structures are or are not detectable in germ cells in adult females. Without further experiments one cannot draw reliable conclusions about neo-oogenesis/de novo meiosis initiation in adult NMRs.

If HORMAD1 staining is unsuitable for the detection of axial elements in sections, and if the authors want to show sections, other markers of the synaptonemal complex would suffice as long as they clearly mark filamentous structures specifically in meiotic prophase. I realize that detection of distinguishing meiotic chromosome structures in chromosome spreads is tedious as only a small proportion of cells are expected to be in meiosis in ovaries of adults. Nonetheless chromosome spreads should be attempted and quantification of examined cells should be provided because axial elements/synaptonemal complex should be detectable in at least some cells if, as hypothesized by the manuscript, de novo meiosis takes place in levels that can meaningfully replenish the oocyte pools in adult NMRs. If no cells are found in meiotic prophase after examining a few thousand ovarian germ cells (this is feasible) the authors would be able to conclude that de novo meiosis in adults is unlikely to contribute to maintenance of fertility in NMRs. These experiments could be carried out by REC8 staining chromosome spreads because in chromosome spreads REC8 patterns are easily distinguishable in prophase cells (axial elements) and oocytes that progressed beyond prophase (no axial elements).

Further specific comments to statements of the manuscript that are not sufficiently supported by data:

(1)

Title: Postnatal oogenesis leads to an exceptionally large ovarian reserve and protracted fertility in naked mole-rats

This title gives the impression that a shift of oogenesis to postnatal stages leads to large ovarian reserve and protracted fertility in naked mole-rats. In conjunction with the rest of the manuscript it also suggests neo-oogenesis in adults without explicitly stating it. The data does not support these conclusions. It is true that oogenesis is postnatal in NMRs but it is not shown or tested that the size of the oocyte reserve and/or the protracted fertility depends on a shift in the developmental timing of oogenesis. Further neo-oogenesis is not convincingly shown in adults. As discussed in the previous two review rounds, the data show only a very moderate shift in the developmental timing of the bulk of meiotic entry (from prenatal to juvenile 15-28 days old NMRs). The size of this shift is very little as compared to the reproductive lifespan and it is not clear how a shift of a few weeks would allow prolongation of fertility to over 30 years.

(2)

Taken together, our results suggest that highly

39 desynchronized germ cell development and the maintenance of a small population of PGCs that

40 can expand upon reproductive activation is a unique strategy for generating a large

ovarian

41 reserve sufficient for the 30-year reproductive lifespan of NMRs.

This part of the abstract suggests that the mentioned characteristics of germ cell differentiation enable long term fertility. But evidence is not provided. Generation of a large ovarian reserve either before (humans) or soon after birth (NMR) is able to support long reproductive lifespan. Hence time does not seem to matter, instead the size of the pool is likely to matter. The paper does not provide conclusive evidence for de novo initiation of meiosis in adults (a stringent criteria for meiosis is the existence of meiotic prophase specific chromosome structures). The authors did not find any cell where chromosome structures that are specific to meiotic prophase could be confidently identified. Accordingly these sentences have to be toned down.

(3)

Collectively, we establish that the entire

94 process of oogenesis occurs postnatally in NMR ovaries, which provides strong support for the

95 hypothesis that pre-meiotic germ cell proliferation throughout the postnatal period generates and

96 maintains an extended ovarian reserve in NMRs.

97

This is overstatement. It is clear from the data that postnatal initiation of meiosis generates a large ovarian reserve, but according to the data it is unlikely that postnatal germ cell proliferation meaningfully helps to maintain the ovarian reserve in adults.

(4)

In P8 and P28 ovaries, we detected all substages of meiotic

272 prophase I (Figure 5d, e, and f). The progression of synapsis was evaluated using antibodies

273 against REC8 and HORMAD1, the latter protein associating only with early (unsynapsed)

274 prophase I oocytes^{56,57} (Figure 5e).

Missleading statement as diplonema cell is shown with HORMAD1 signal in 5d. Diplonema is in late prophase.

(5)

Collectively, these results show that ovaries from reproductively active NMRs have a population

358 of germ cells that are able to divide in vivo and in vitro and enter into meiotic prophase I, 359 suggesting they are capable of contributing to the maintenance of the NMR ovarian reserve

This message needs to be toned down. Provided that, in future experiments, no oocytes are found with axial elements/synaptonemal complex in adults, it needs to be stated that no evidence for functionally normal meiosis is found and that the low proportions of HORMAD1 cells is a warning sign that cells dividing late and attempting to enter meiosis may fail in oogenesis.

(6)

Whereas we

244 observed REC8 signaling in follicles as expected, we excluded REC8+ follicles from the counts,

245 because we wanted to quantify only those oocytes during meiotic prophase before the dictyate

246 arrest and follicle development.

I guess "REC8 signal" should replace "REC8 signaling" as, to my knowledge, REC8 is not involved in signaling pathways.

(8)

Fig 8 legend.

I may have missed it, but I did not find reference to the arrows in figure 8a, c, e.

REVIEWERS' COMMENTS

Reviewer #3 (Remarks to the Author):

R3.Q1. The main conclusions of the manuscript are that (i) both mitotic germ cell proliferation and meiosis entry postnatally take place in naked mole rats (NMRs), and that (ii) de novo meiosis entry/neo-oogenesis occurs in adults, which is a unique reproductive strategy that supports maintenance of oocyte pools and fertility during the long reproductive life of NMRs. The previous version of the manuscript convincingly showed that a large cohort of germ cells enters meiosis soon after birth thereby generating a large oocyte reserve in juvenile NMRs. However, evidence was missing for de novo meiosis initiation in adults. The new version of manuscript includes new data that aims to fill this gap. To this end marker proteins of meiotic prophase, STRA8, gamma H2AX and HORMAD1, were detected in sections of ovaries of adult NMRs (note that “adult” refers to SUB 3yr and EX-SUB 3yr in this review).

We want to thank the reviewer for the comments to improve our work. As reviewer #3 mentioned, our experiments were focused on the possibility of germ cells entering meiotic prophase in adults. Our results showed that reproductive activation of naked mole-rats at 3 years of age induces a statistically significant increase in the number of germ cells that are mitotically active, as well as the number of germ cells that are positive for markers of meiotic prophase initiation (STRA8, γ H2AX and, HORMAD1).

R3.Q2. Whereas these experiments formally attempted to satisfy requests from the previous review round, they fell short of addressing the essence of the requests. Hence, the main problematic issue remains the same, de novo initiation of normal meiosis is not conclusively demonstrated. In the previous review round, it was requested that a marker of meiosis initiation is detected in germ cells of adults to test if meiosis initiation takes place. HORMAD1 was suggested as a reliable marker and, in general, the use chromosome spreads was encouraged, because unique meiosis specific chromatin structures, axial elements and the synaptonemal complexes, which are marked by HORMAD1 or REC8 are much easier detected on nuclear surface spreads as opposed to sections. It is very dangerous to rely only on expression of marker proteins as opposed to their subcellular localization as a criteria for judging if meiosis is induced in germ cells. In the past, a reliance on protein and RNA expression data led to the meanwhile discredited notion of oogenic stem cells/meiosis in adult in humans. Therefore, a detection of chromosome structures that are unique to meiosis are essential to test if a functional meiosis is initiated in adults. The presented data does not reveal if meiosis specific chromosome structures normally form in germ cells of adult NMRs. The granular/pan-nuclear HORMAD1 staining patterns are surprising, as filamentous structures resembling REC8 patterns (fig 5) would be expected for an axial element component, such as HORMAD1, in meiotic prophase. The HORMAD1 staining patterns may have two possible explanations. (i) HORMAD1 is not localized on axial elements in oocytes of adult NMRs. Hence, the HORMAD1 staining shown in figure 8 may represent potentially abnormal expression of HORMAD1 in non-meiotic cells. (ii) Nucleoplasmic HORMAD1 may mask filamentous HORMAD1 patterns in functionally normal meiotic cells in sections. To distinguish between these possibilities HORMAD1 should have been detected in chromosome spreads in oocytes of 3 years old NMRs as suggested in the previous review round. Further, as a control, HORMAD1 could have been detected in perinatal (15-28 days) sections to test if HORMAD1 staining produces filamentous patterns in sections at a stage where large fractions of oocytes enter meiosis. I note that It could not be predicted in advance if HORMAD1 staining on sections would or would not reproduce the filamentous patterns that are typical of axial element components such as REC8 (see figure 5, both sections and nuclear surface spreads). Therefore,

the use of both chromosome spreads (for reliable detection of axial elements) and sections (for simplicity and reduced effort) were suggested in the previous review round.

We agree with reviewer #3 that subcellular localization of marker proteins is a better criterion for judging whether meiosis is induced in germ cells. We also agree that in the past, a reliance on protein and RNA expression data led to the since discredited notion of oogenic stem cells/meiosis in adult humans. However, in our manuscript we used three different markers to detect initiation of meiotic prophase I. As Reviewer#3 noted previously, we performed IF analyses with the HORMAD1 antibody on tissue sections from young animals (P8, P28 and P90). In fact, results from these tests are now added as a supplementary figure 10. The analyses of these sections revealed that cells were positive for nuclear HORMAD1 in an identical manner as in adults (nuclear signal without filamentous structure). As the reviewer clearly indicated, detection of filamentous structures within these cells using antibodies against HORMAD1 or other synaptonemal markers would be the preferred method to demonstrate that these cells not only trigger the meiotic program, but also progress through meiotic prophase. The Briño-Enríquez lab is actively pursuing this approach, and it is our goal in future experiments to show the presence of axial elements/synaptonemal complex proteins, as well as meiotic recombination profiles in the adult NMRs from different ages. Whereas these sorts of experiments are among our long-term research goals on NMR ovarian biology, they will require several years to complete and would constitute an entirely new project.

R3.Q3. A lack of analysis of subcellular localization of HORMAD1 is not the only problem. It is very worrying that HORMAD1 is present in much lower numbers (around 300 cell) of VASA positive germ cells than STRA8 or gamma H2AX (both are in the 7000 range, 20-25 fold more), which are also used as meiotic markers (fig. 8). Based on the developmental expression patterns of these markers in mice, HORMAD1 should be expressed in similar or possibly higher proportions of meiotic cells than STRA8 or gammaH2AX. The discrepancies in the frequency of meiotic markers suggest that a functional meiotic programme is inefficiently, if at all, initiated in adult NMRs. Strong gamma H2AX accumulation can be observed also in meiotic cells that undergo apoptosis in mice, hence the data may indicate that germ cells attempt to initiate meiosis but fail and undergo cell death in adults. These considerations also highlight the dangers of relying on expression of marker proteins as opposed to their subcellular localization as a criteria for meiosis. In short, in my opinion, the presented data argue against the hypothesis that functionally normal meiosis takes place in significant numbers in adult NMRs. At best, the data are inconclusive about meiosis in adult female NMRs. Therefore, further data should be included to clarify if meiosis specific chromosome structures are or are not detectable in germ cells in adult females. Without further experiments one cannot draw reliable conclusions about neo-oogenesis/de novo meiosis initiation in adult NMRs.

We understand Reviewer #3 concerns about the apparent discrepancy between the greater number of cells labeled with STRA8 and γ H2AX as compared to those stained with HORMAD1. While we were ourselves not expecting to see such a difference, it is not altogether surprising, for a number of reasons:

- a) The relative abundance and/or intensity of staining for γ H2AX vs HORMAD1 is likely to be quite different, and this could be reflected by different sensitivities in IF staining
- b) The progression of zygonema is very dynamic, encompassing cells with a high degree of asynapsis (early zygonema) right through to those cells with a very high degree of synapsis (late zygonema). While γ H2AX signal would continue throughout

this dynamic progression, **HORMAD1** staining would be limited only to those cells showing higher levels of asynapsis, and thus **HORMAD1** staining would be more restrained in late zygonema.

- c) The level and pattern of **HORMAD1** staining is inferred from studies of male meiosis in mice and thus could be very different in female NMRs.

For these, and other, reasons, we find the **HORMAD1** and **gH2AX** staining differences to be interesting, but not concerning. We agree that without further analyses we cannot say anything about whether these cells with an active meiotic program will be functional or not. Nonetheless, the presence of both these markers and **STRA8** provide convincing evidence for initiation of de novo meiosis in adult NMR females.

R3.Q4. If **HORMAD1** staining is unsuitable for the detection of axial elements in sections, and if the authors want to show sections, other markers of the synaptonemal complex would suffice as long as they clearly mark filamentous structures specifically in meiotic prophase. I realize that detection of distinguishing meiotic chromosome structures in chromosome spreads is tedious as only a small proportion of cells are expected to be in meiosis in ovaries of adults. Nonetheless chromosome spreads should be attempted and quantification of examined cells should be provided because axial elements/synaptonemal complex should be detectable in at least some cells if, as hypothesized by the manuscript, de novo meiosis takes place in levels that can meaningfully replenish the oocyte pools in adult NMRs. If no cells are found in meiotic prophase after examining a few thousand ovarian germ cells (this is feasible) the authors would be able to conclude that de novo meiosis in adults is unlikely to contribute to maintenance of fertility in NMRs. These experiments could be carried out by **REC8** staining chromosome spreads because in chromosome spreads **REC8** patterns are easily distinguishable in prophase cells (axial elements) and oocytes that progressed beyond prophase (no axial elements).

As mentioned in the manuscript, the different substages of meiotic prophase I are typically evaluated according to the characteristics of the synaptonemal complex. However, the antibodies for these proteins (**SYCP2**, **SYCP3**) used in mice produced inadequate results in NMRs; therefore, we were unable to use them on tissue sections. As mentioned in response to **R3.Q2**, the Briño-Enríquez lab is actively working on this topic, and it is our goal in future experiments to show the presence of axial elements/synaptonemal complex and meiotic recombination profiles using chromosome spreads in the adult NMRs of different ages. We are developing new techniques and antibodies that will help us to analyze the full meiotic prophase in older adult NMRs. Whereas these sorts of experiments are among our long-term research goals on NMR ovarian biology, they will require several years to complete.

Further specific comments to statements of the manuscript that are not sufficiently supported by data:

R3.Q5. Title: Postnatal oogenesis leads to an exceptionally large ovarian reserve and protracted fertility in naked mole-rats

This title gives the impression that a shift of oogenesis to postnatal stages leads to large ovarian reserve and protracted fertility in naked mole-rats. In conjunction with the rest of the manuscript it also suggests neo-oogenesis in adults without explicitly stating it. The data does not support these conclusions. It is true that oogenesis is postnatal in NMRs but it is not shown or tested that the size of the oocyte reserve and/or the protracted fertility depends on a shift in the

developmental timing of oogenesis. Further neo-oogenesis is not convincingly shown in adults. As discussed in the previous two review rounds, the data show only a very moderate shift in the developmental timing of the bulk of meiotic entry (from prenatal to juvenile 15-28 days old NMRs). The size of this shift is very little as compared to the reproductive lifespan and it is not clear how a shift of a few weeks would allow prolongation of fertility to over 30 years.

Within the manuscript, we deliberately stated only that oogenesis occurs postnatally, while never using the term “neo-oogenesis”. Therefore, even when the greatest number of germ cells are generated during first month of postnatal development, it results in the generation of exceptionally large number of germ cells relative to the NMR’s diminutive body size, with a low level of apoptosis/atresia.

R3.Q6. Taken together, our results suggest that highly desynchronized germ cell development and the maintenance of a small population of PGCs that can expand upon reproductive activation is a unique strategy for generating a large ovarian reserve sufficient for the 30-year reproductive lifespan of NMRs.

This part of the abstract suggests that the mentioned characteristics of germ cell differentiation enable long term fertility. But evidence is not provided. Generation of a large ovarian reserve either before (humans) or soon after birth (NMR) is able to support long reproductive lifespan. Hence time does not seem to matter, instead the size of the pool is likely to matter. The paper does not provide conclusive evidence for de novo initiation of meiosis in adults (a stringent criteria for meiosis is the existence of meiotic prophase specific chromosome structures). The authors did not find any cell where chromosome structures that are specific to meiotic prophase could be confidently identified. Accordingly these sentences have to be toned down.

We modified the sentence to “Collectively, our results suggest that highly desynchronized germ cell development and the maintenance of a small population of PGCs that can expand upon reproductive activation are unique strategies that could help to maintain the NMR’s ovarian reserve for its 30-year reproductive lifespan” (Lines 34-37).

R3.Q7. Collectively, we establish that the entire process of oogenesis occurs postnatally in NMR ovaries, which provides strong support for the hypothesis that pre-meiotic germ cell proliferation throughout the postnatal period generates and maintains an extended ovarian reserve in NMRs.

This is overstatement. It is clear from the data that postnatal initiation of meiosis generates a large ovarian reserve, but according to the data it is unlikely that postnatal germ cell proliferation meaningfully helps to maintain the ovarian reserve in adults.

We modified the sentence to “Collectively, we establish that the entire process of oogenesis occurs postnatally in NMR ovaries, which provides strong support for the hypothesis that pre-meiotic germ cell proliferation throughout the postnatal period generates an exceptionally large ovarian reserve relative to the NMR’s body size” (lines 93-96)

R3.Q8. In P8 and P28 ovaries, we detected all substages of meiotic prophase I (Figure 5d, e, and f). The progression of synapsis was evaluated using antibodies against REC8 and HORMAD1, the latter protein associating only with early (unsynapsed) prophase I oocytes^{56,57} (Figure 5e). Misleading statement as diplonema cell is shown with HORMAD1 signal in 5d. Diplonema is in late prophase.

The sentence was modified to “In P8 and P28 ovaries, we detected all substages of meiotic prophase I (Figure 5d, e, and f). The progression of synapsis was evaluated using antibodies against REC8 and HORMAD1, the latter protein associating only with unsynapsed regions in prophase I oocytes^{56,57} (Figure 6e)”. (Lines 277-279)

R3.Q9. Collectively, these results show that ovaries from reproductively active NMRs have a population of germ cells that are able to divide *in vivo* and *in vitro* and enter into meiotic prophase I, suggesting they are capable of contributing to the maintenance of the NMR ovarian reserve

This message needs to be toned down. Provided that, in future experiments, no oocytes are found with axial elements/synaptonemal complex in adults, it needs to be stated that no evidence for functionally normal meiosis is found and that the low proportions of HORMAD1 cells is a warning sign that cells dividing late and attempting to enter meiosis may fail in oogenesis.

We toned down and modified the sentence to: “Collectively, these results show that ovaries from reproductively active NMRs have a population of germ cells that are able to divide *in vivo* and *in vitro* and trigger meiotic prophase I, suggesting they could be capable of contributing to the maintenance of the NMR ovarian reserve. However, more studies are warranted to determine whether complete meiotic prophase progression and follicle formation occurs in adult NMRs”. (Lines 367-371)

R3.Q10. Whereas we observed REC8 signaling in follicles as expected, we excluded REC8+ follicles from the counts, because we wanted to quantify only those oocytes during meiotic prophase before the dictyate arrest and follicle development.
I guess “REC8 signal” should replace “REC8 signaling” as, to my knowledge, REC8 is not involved in signaling pathways.

We changed the sentence to “REC8 signal” as Reviewer#3 suggested. (lines 248-249)

R3.Q11. Fig 8 legend. I may have missed it, but I did not find reference to the arrows in figure 8a, c, e.

The figure legend (old figure 8, new figure 9) now states “white arrows indicate representative positive cells”. Line 1040